# SeCon-RAG: A Two-Stage Semantic Filtering and Conflict-Free Framework for Trustworthy RAG

Xiaonan Si[1*], Meilin Zhu[2,3*], Simeng Qin[4†], Lijia Yu[5†], Lijun Zhang[1†], Shuaitong Liu[6], Xinfeng Li[7],
Ranjie Duan[8], Yang Liu[7], Xiaojun Jia[7†]

[1]Institute of Software, Chinese Academy of Sciences, Beijing, China
[2]Key Laboratory of System Software (Chinese Academy of Sciences) and State Key Laboratory of
Computer Science, Institute of Software, Chinese Academy of Sciences, Beijing, China
[3]University of Chinese Academy of Sciences, Beijing, China
[4]Northeast University, China      [5]Institute of Ai For industries, Nanjing, China
[6]Southwest University, China      [7]Nanyang Technological University, Singapore      [8]Alibaba, China

## Abstract

Retrieval-augmented generation (RAG) systems enhance large language models (LLMs) with external knowledge but are vulnerable to corpus poisoning and contamination attacks, which can compromise output integrity. Existing defenses often apply aggressive filtering, leading to unnecessary loss of valuable information and reduced reliability in generation. To address this problem, we propose a two-stage semantic filtering and conflict-free framework for trustworthy RAG. In the first stage, we perform a joint filter with semantic and cluster-based filtering which is guided by the Entity-intent-relation extractor (EIRE). EIRE extracts entities, latent objectives, and entity relations from both the user query and filtered documents, scores their semantic relevance, and selectively adds valuable documents into the clean retrieval database. In the second stage, we proposed an EIRE-guided conflict-aware filtering module, which analyzes semantic consistency between the query, candidate answers, and retrieved knowledge before final answer generation, filtering out internal and external contradictions that could mislead the model. Through this two-stage process, SeCon-RAG effectively preserves useful knowledge while mitigating conflict contamination, achieving significant improvements in both generation robustness and output trustworthiness. Extensive experiments across various LLMs and datasets demonstrate that the proposed SeCon-RAG markedly outperforms state-of-the-art defense methods.

## 1 Introduction

Large Language Models (LLMs) [5, 12, 32] have demonstrated remarkable capabilities across a wide range of natural language tasks [51, 15, 14]. However, they still suffer from critical security vulnerabilities, including adversarial attacks [48, 29], jailbreak attacks [21, 19, 46], and other alignment challenges. Moreover, their knowledge is fundamentally limited by their training data, which can lead to outdated or hallucinated information. Retrieval-Augmented Generation (RAG) addresses this issue by dynamically incorporating external documents during generation, improving factual accuracy and timeliness [25, 3]. However, due to the reliance on external corpora, RAG systems are susceptible to corpus poisoning and retrieval contamination attacks, which involve injecting adversarial content into the retrieval database to manipulate the model's output [31, 49, 7].

---

*These authors contributed equally. The code is available `here`.
†Corresponding authors: Simeng Qin( qinsimeng@neuq.edu.cn), Lijia Yu (ljyu@iaii.ac.cn), Lijun Zhang (zhanglj@ios.ac.cn), Xiaojun Jia (jiaxiaojunqaq@gmail.com)

39th Conference on Neural Information Processing Systems (NeurIPS 2025).

Recent defense strategies have attempted to address this by employing adversarial training, retrieval filtering and reasoning-based conflict resolution [44, 39, 54]. These methods primarily use Coarse-grained filtering or voting to remove malicious documents, and the inference phase does not consider what information the RAG should select when confronted with conflicting content, which can result in two limitations. (1) Coarse-Grained filtering will removes both harmful and useful content. (2) Failure to resolve conflicts between retrieved and the LLM's internal knowledge, which leads to untrustworthy results.

To address these issues, our framework first integrates semantic information into the RAG filtering method. We extract intrinsic semantic signals from each document to allow for fine-grained filtering while also facilitating the resolution of conflicting evidence during inference. Building on this insight, we propose SeCon-RAG, a two-stage framework that combines semantic and cluster-based filtering with conflict-filtering retrieval-augmented generation.

We first design a semantic extraction module called EIRE (Entity-Intent-Relation Extractor). It makes future modules easier to use by extracting entities, hidden intentions, and relationships between entities from document information. In the first stage, we propose a Semantic and Clustering-Based Filtering module (SCF) based on EIRE. On the one hand, it filters the intensive incorrect documents based on their cluster in the embedding space. On the other hand, using EIRE, the semantic structure graph of candidate documents and verified correct documents can help to exclude more hidden poisoned documents. The implementation of this dual filtering mechanism can ensure that the majority of malicious and poisonous documents are filtered out while also preventing potentially valuable documents from being wasted.

In the second stage, we propose an EIRE-guided conflict-aware filtering (CAF) module that checks the semantic consistency of the query, the candidate context, and the model's internal knowledge. CAF uses EIRE to extract semantic information from the final input information, judge different information based on semantic knowledge, and remove misleading information caused by internal and external knowledge conflicts or omissions before generating the final response. This ensures that the final generations are not only factually accurate, but also semantically consistent across internal and external knowledge sources.

In comparison to previous work, our work makes significant advances. Our approach is the first to incorporate semantic information into the retrieval and inference phases of RAG defenses. We propose a two-stage defense framework that employs semantic reasoning to ensure robust during retrieval (SCF) and generation (CAF). The proposed framework implements structured semantic filtering by extracting entity-intent relationships and using them to filter poisoning documents which may evade clustering-based defenses.

We evaluate SeCon-RAG on three QA benchmarks Natural Questions, HotpotQA, and MS-MARCO across five different LLMs including LLaMA-3.1-8B [13], Mistral-12B [2], GPT-4o [1], DeepSeek-R1 [18], and Qwen-7B [20]. Our method consistently improves robustness, consistency, and resistance to corpus poisoning across all settings. Our main contributions are summarized as follows:

(a) We are the first to incorporate structured semantic information into RAG defense filtering by the proposed EIRE module, allowing for fine-grained understanding of entity, intent, and relation structures to improve the precision of poisoned content detection. (b) Building on EIRE, we propose SeConRAG, a two-stage defense framework that combines semantic and cluster-based filtering with conflict-aware filtering to improve retrieval robustness and answer consistency. (c) Extensive experiments on a variety of datasets and LLMs show that SeConRAG consistently achieves high factual accuracy, low attack success rates, and high generalizability, demonstrating its practical effectiveness and plug-and-play capabilities.

## 2 Related Works

### 2.1 Retrieval-Augmented Generation

Retrieval-Augmented Generation improves large language models by supplementing them with external knowledge extracted from large corpora, thereby addressing limitations in factual recall and knowledge coverage [25, 43]. While RAG's generation quality has improved, it continues to suffer from retrieval errors, hallucinations, and poor content integration. To address these issues,

previous research has focused on query rewriting, index optimization, and memory-based retrieval [52, 28]. Recent LLM-augmented methods include Insight-RAG [33], SURE [23], and PIKE-RAG [40], which use LLMs to improve task comprehension, retrieval relevance, and data decomposition [34]. Reinforcement learning has also been applied to optimize retrieval generation pipelines [50]. However, these methods are primarily applicable in benign environments and do not explicitly address poisoning threats or semantic inconsistencies caused by conflicting retrieved content.

## 2.2 Adversarial Attacks on RAG

Recent research indicates that RAG systems are extremely vulnerable to adversarial manipulation at both the input and corpus levels. Attack strategies include: (1) Corpus Poisoning Attacks, which inject adversarially crafted documents into the retrieval corpus and manipulate downstream outputs [49, 37, 55, 36, 9, 45, 53, 31]. (2) Prompt Injection Attacks, which use imperceptible instructions embedded in user queries or retrieved content to hijack LLM behavior without altering the underlying corpus [35, 22, 26]. (3) Backdoor Attacks, in which hidden triggers are implanted into the corpus or model and activated only under certain conditions [27, 10]. These attacks destroy the reliability of RAG outputs and expose the system to silent failure scenarios.

## 2.3 Defenses Against poisoning RAG

A variety of defense strategies have been proposed to counter adversarial threats. Perplexity-based detectors seek to identify anomalous generations, whereas RevPRAG examines LLM activation patterns to detect poisoned inputs [36, 38]. RobustRAG introduces an isolate-then-aggregate framework to improve robustness by decoupling retrieval paths, while AstuteRAG adaptively fuses internal knowledge with retrieved content using heuristic selection [44, 39]. InstructRAG enhances Retrieval-Augmented Generation by employing self-synthesized rationales, guiding the retrieval process to improve the relevance and coherence of generated outputs [42]. TrustRAG filters out malicious content using clustering over document embeddings and introduces a conflict resolution mechanism based on document consistency [54]. Although promising, these approaches have two major limitations: Majority-voting often fails under high poisoning, while heuristic and aggressive filtering may lose relevant content under low poisoning.

In contrast to previous work, we propose SeCon-RAG, a robust two-stage framework for fine-grained semantic filtering and conflict-aware inference. SeCon-RAG improves robust against both high and low poisoning setting by leveraging intrinsic semantic signals and reasoning over document-level consistency, while preserving valuable information for reliable generation.

## 3 Preliminary

This section provides a brief overview of Retrieval-Augmented Generation and introduces the threat model of corpus poisoning attacks that underpins the defense strategies proposed in this paper.

### 3.1 Retrieval-Augmented Generation

Retrieval-Augmented Generation is a widely used paradigm for augmenting large language models with external knowledge obtained from a document corpus. Given a user query $q$ and a corpus $\mathcal{D} = \{d_i\}$, where $d_i$ represent the documents in $\mathcal{D}$. The standard RAG framework has three primary stages. In the first stage, compress the query $q$ and the documents $d_i$ in $\mathcal{D}$ into $E(q)$ and $E(d_i)$ using the embedding model $E$. In the second stage, select the top-k documents with the highest similarity to the problem in the document to form a set $\mathcal{D}_k(q)$. The similarity is determined by a given function $\text{sim}(\cdot, \cdot)$, as follows:

$$\mathcal{D}_k(q) = \text{Top-k}_{d \in \mathcal{D}}\{\text{sim}(E(q), E(d))\}, \tag{1}$$

Finally, the retrieved documents $D_k(q)$ are combined with the original query $q$ to create an augmented input prompt. The augmented input is processed by a generative model $F$, such as a large language model, i.e. $F(q, \mathcal{D}_k(q))$, to generate the final output.

## 3.2 Threat Model: Corpus Poisoning Attacks

We examine a threat model that tries to trick a RAG system into producing incorrect answers by inserting carefully crafted malicious documents into its retrieval corpus. The attacker chooses $M$ target queries $\mathcal{Q} = \{q_1, q_2, \ldots, q_M\}$ and matches each query $q_i$ with a poisoning target answer $r_i$. For example, for $q_i$ = "Who is the president of America?", the adversary may want the RAG system to produce $r_i$ = "The president of America is Harris" . To achieve this, the attacker injects $N$ poisoning documents per query. Let $p_i^j$ denote the $j$-th poisoned document for query $q_i$, where $j = 1, \ldots, N$. The total set of injected documents is:

$$\Gamma = \{p_i^j \mid i = 1, \ldots, M; \ j = 1, \ldots, N\} \tag{2}$$

The attack aims to create $\Gamma$ so that, for each query $q_i \in \mathcal{Q}$, RAG system retrieves documents from the poisoned corpus $\mathcal{D}' = \mathcal{D} \cup \Gamma$ that lead the generative model $F$ to produce the incorrect response $r_i$:

$$F(q_i, \mathcal{D}'_k(q_i)) \approx r_i, \ \forall i \in [M]. \tag{3}$$

This threat model is consistent with previous research on corpus poisoning and informs our design of a filtering-based defense strategy. In the following sections, we present our proposed SeCon-RAG framework, which combines two-stage filtering to protect against corpus poisoning attacks.

# 4 The Proposed Defense Method for Corpus Poisoning Attacks

To protect Retrieval-Augmented Generation systems from corpus poisoning attacks, we propose SeCon-RAG, a robust two-stage filtering framework designed to detect and suppress poisoning documents. The first stage eliminates poisoned content statistically and semantically, while the second stage ensures factual consistency from a semantic reasoning perspective. This design ensures robustness without unnecessary knowledge loss. To enable fine-grained semantic understanding and aid in the detection of potentially poisoned content, we propose Entity-Intent-Relation Extractor in section 4.1, a semantic structure extraction module that serves as the foundation for our two-stage filtering framework. Before the retrieval stage, we propose Semantic and Cluster-Based Filtering shown in section 4.2 creates a semantic graph from the information extracted by EIRE, allowing for dual-channel filtering based on both clustering structure and semantic relevance. During the inference stage, we introduce the Conflict-Aware Filtering module shown in section 4.3. CAF performs cross-source semantic consistency checks using both EIRE on the retrieved content and the model's internal knowledge representations. Figure 1 shows an overview of the full Secon-RAG framework. The appendix A.3 shows the pseudocode for the overall algorithm.

## 4.1 EIRE: Entity-Intent-Relation Extractor

To enable fine-grained semantic understanding and aid in the detection of potentially poisoned content, we propose EIRE (Entity-Intent-Relation Extractor), a semantic structure extraction module that serves as the foundation for our two-stage filtering framework. EIRE is intended to capture the high-level meaning of a document by breaking it down into three core structural components:

- **Entities**: Key entities explicitly or implicitly mentioned in the text.
- **Intent**: The underlying purpose or objective conveyed by the passage.
- **Relations**: Semantic relationships between extracted entities, such as *beat* or *followed by*.

To extract these components, EIRE employs a prompt-based large language model. Given a document $d$, we create structured prompts that direct the LLM to generate a structured triple. Appendix A.1 contains an example of the prompt and its output. For a document $d$, EIRE generates a structured triple $(E_d, I_d, R_d)$, where $E_d$ is the set of extracted entities, $I_d$ is the identified intent, and $R_d$ is the set of semantic relations between entities. By grounding document analysis in interpretable semantic frames, EIRE provides a robust and explainable foundation for downstream filtering.

## 4.2 Semantic and Clustering-Based Filtering

To reduce the risk of retrieving poisoned or adversarial documents, we introduce a dual filtering mechanism in the retrieval stage called Semantic and Clustering-Based Filtering (SCF). SCF is applied before selecting $\mathcal{D}_k(q)$ in the RAG pipeline.

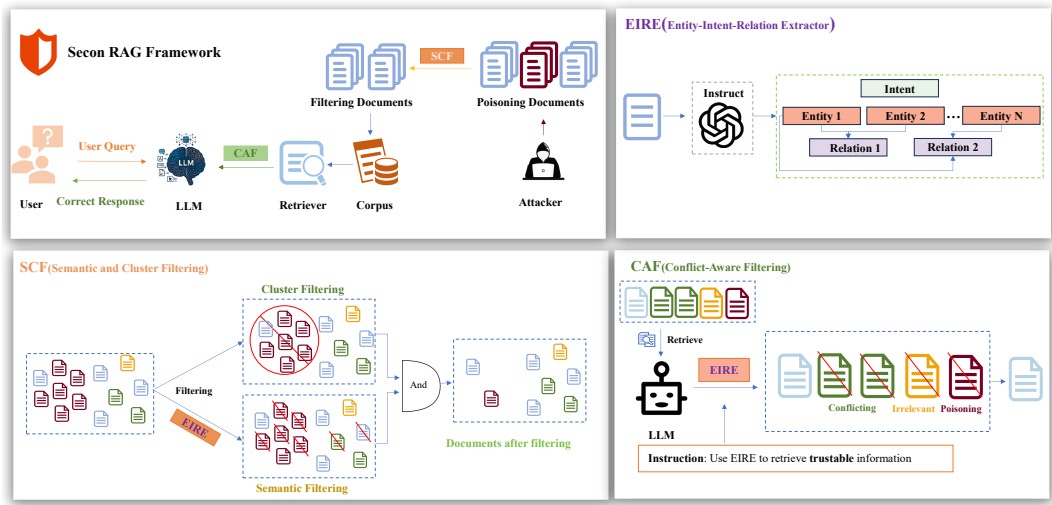

Figure 1: Overview of the SeCon-RAG. A two-stage defense in which SCF filters poisoning corpus during retrieval and CAF eliminates residual conflicts during inference, guided by semantic information obtained through EIRE.

### 4.2.1 Clustering-Based Filtering

Adversarially generated poisoning documents often exhibit highly similar phrasing or templated structures, especially when crafted to target the same query. As a result, they naturally form tight clusters in the embedding space [54]. To mitigate this, we first apply a clustering-based filter to detect poisoning document groups. Given a potentially poisoned corpus $\mathcal{D}' = \mathcal{D} \cup \Gamma$, we embed each document $d \in \mathcal{D}'$ into the vector representation $m(d)$ and apply K-means clustering to obtain $K$ clusters $C = \{c_1, \ldots, c_K\}$, each with centroid $\mu_i = \frac{1}{|c_i|} \sum_{d_j \in c_i} m(d_j)$, $\bigcup_{i=1}^{K} c_i = \mathcal{D}'$ [30]. We then define the filtered set as:

$$\mathcal{D}_{\text{cluster}} = \bigcup_{i=1}^{K} \{d \in c_i \mid \text{sim}(m(d), \mu_i) \leq \tau_{\text{cluster}}\} \tag{4}$$

where $sim(\cdot, \cdot)$ denotes the cosine similarity normalized to $[0, 1]$, and $\tau_{\text{cluster}} \in (0, 1)$ is an adjustable filtering threshold. This operation effectively exclude documents that cluster too tightly around a centroid, which are likely to be maliciously inserted poisoning documents.

### 4.2.2 Semantic Graph-Based Filtering by EIRE

However, clustering-based methods rely solely on vector similarity in the embedding space, which can lead to false negatives by discarding valuable documents like topic overlap. To address this, we propose a semantic filter based on EIRE that extracts semantic structures from individual documents and generates corresponding semantic graphs. Specifically, for a document $d$, we construct a semantic relevance graph $G_d = (V_i, E_{ij})$ by using information extracted from EIRE to simulate the semantic coherence and connectivity of the document $d$ as follows:

- $V_i$: Each node $v_i \in V_i$ in the $G_d$ corresponds to the embedding representation of an **entity** extracted from document $d$;
- $E_{ij}$: An edge $e_{ij}$ between two nodes $v_i, v_j$ denotes a semantic **relation** extracted by EIRE connecting the two entities.

Figure 2 visualizes semantic graphs generated by EIRE for correct and poisoned documents under the query: *"Which French ace pilot and adventurer flew L'Oiseau Blanc?"* . As demonstrated, correct documents produce densely connected semantic graphs with high coherence, whereas poisoned documents have sparse or fragmented structures. From a graph-theoretic perspective, the correct document displays a densely connected semantic graph, with the correct answer node well integrated into the EIRE conceptual structure, resulting in semantic graphs with high structural connectivity

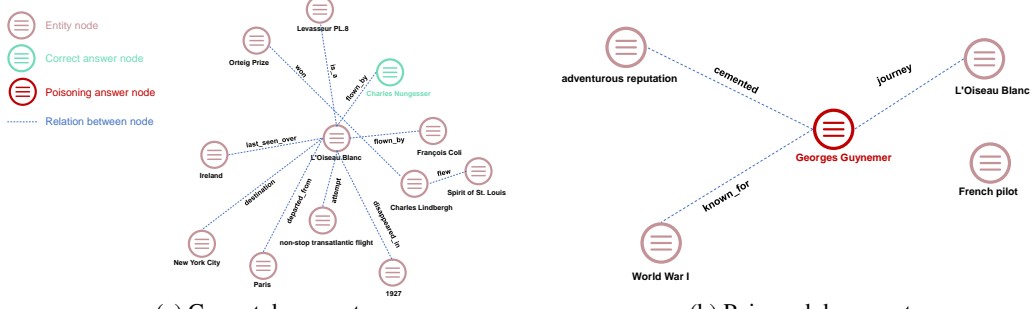

(a) Correct document      (b) Poisoned document

Figure 2: Semantic graph comparison using EIRE, more textual details has shown in AppendixA.1.2.

and semantic coherence. In contrast, poisoning content often introduces isolated or deceptive claims lacking semantic support from the surrounding context. Consequently, their semantic graphs exhibit abrupt or unnatural connections, with isolated nodes or disconnected subgraphs, in sharp contrast to the coherent clusters in correct documents. This structural distinction underpins our semantic filtering strategy.

To use these structural properties, we first construct a set of semantic graphs $G_{\text{cor}} = \{G_{d_{\text{cor},i}}\}$ from a small collection of verified correct documents $D_{\text{cor}} = \{d_{\text{cor},i}\}$ and use the semantic graphs $G_{\text{cor}}$ as a benchmark. $G_{\text{cor}}$ used for semantic reference are a small set of samples chosen manually from the dataset. For any candidate document $d \in \mathcal{D}'$, we generate its semantic graph $G_d$ using EIRE and assess its similarity to the $G_{\text{cor}}$. Rather than relying on rigid graph similarity metrics, we employ large language models' semantic reasoning capabilities to compare graph structures in a more flexible and context-aware way. For any candidate document $d$, we compute its semantic similarity score $ssG$ by comparing generate $G_d$ to $G_{d_{cor,i}}$ using a prompt-based LLM as shown in appendix A.1:

$$ssG(d, D_{\text{cor}}) = \text{LLM}(G_d, G_{\text{cor}}) \tag{5}$$

To facilitate downstream filtering, we limit the LLM-derived semantic similarity score $ssG(d, D_{\text{cor}}) \in [0, 1]$. The higher the similarity score, the closer the semantic graph of $d$ and baseline $G_{\text{cor}}$. Using this score, we define the semantically filtered document set as:

$$\mathcal{D}_{\text{semantic}} = \{d \in \mathcal{D}' \mid ssG(d, D_{\text{cor}}) \leq \tau_{\text{semantic}}\} \tag{6}$$

where $\tau_{\text{semantic}}$ is the adjustable threshold that controls the strictness of semantic filtering. It is worth noting that, while vector projections are used to visualize and shape semantic graphs, the real inputs to the LLM in Equation (5) are natural language descriptions of the graphs serialized as structured triples.

### 4.2.3 Joint Filtering Decision: Robust AND Logic

To increase robustness while reducing the risk of discarding valuable information, we use a conservative AND-based filtering strategy. Only documents that have been flagged by both clustering and semantic filters are filtered. We define the final set of filtered documents as $\mathcal{D}_{\text{final}} = \mathcal{D}_{\text{cluster}} \cap \mathcal{D}_{\text{semantic}}$. Accordingly, the final retained corpus is:

$$\widetilde{\mathcal{D}} = \mathcal{D}' \setminus \mathcal{D}_{\text{final}} \tag{7}$$

By the joint filter, the SCF module combines unsupervised clustering and semantic reasoning to detect poisoning documents from multiple perspectives. This layered approach improves the quality of retrieved documents and provides a robust first line of defense in the SeCon-RAG framework.

### 4.3 Conflict-Aware Filtering (CAF)

Although the SCF module effectively reduces adversarial content, it may retain documents that are not malicious but semantically irrelevant or internally inconsistent. These residual conflicts, such as documents that contradict the query, other retrieved evidence, or the model's internal knowledge, can reduce the factual reliability of the final answer. To address this limitation, we propose Conflict-Aware Filtering (CAF), a semantic inference module used at the inference stage of the RAG. CAF aims to refine the retrieved set $\mathcal{D}_k(q)$ by identifying and removing documents that don't meet semantic and factual consistency criteria.

For each candidate document $d \in \mathcal{D}_k(q)$, CAF generates structured semantic information using EIRE, which is divided into three components: **Entities** capable of determining whether facts align with the model's internal knowledge; **Intent** to evaluate query alignment; **Relations** that can evaluate logical coherence across documents and detect contradictions or omissions. In the final inference process, we prompt the LLMs as shown in appendix A.1 to determine which information from the retrieve documents is reliable from three dimensions using the semantic information extracted by EIRE:

- **Q(Query Consistency)**: Does the document semantically aligned with the user query $q$, based on intent and entities?

- **C(Corpus Consistency)**: Is the document consistent with the other retrieved documents $\mathcal{D}_k(q) \setminus d$, based on shared relations and context?

- **M(Model Consistency)**: Is the document factually compatible with the LLM's internal knowledge, considering key entities?

The LLMs will render a judgment on whether the information is poisoned, conflicting, irrelevant, or trustworthy, which will be used to make final decisions. Finally, the LLMs extracts the final answer $A(q)$ to query $q$ from documents that perform well in these three dimensions:

$$\begin{aligned} A(q) &= F(q, \widetilde{\mathcal{D}}_{\text{CAF}}) \\ \widetilde{\mathcal{D}}_{\text{CAF}} &= \{d \in \mathcal{D}_k(q) \mid \text{CAF}(d, Q, C, M) = \texttt{trustable}\} \end{aligned}$$

Figure 1 shows the operation of CAF. Following filtering by SCF, each document is evaluated by the LLM based on EIRE-derived semantic structure. Information from documents have been identified as poisoned, conflicting, or irrelevant is discarded, leaving only trustable information for final generation. This ensures that the generation module operates on a semantically coherent, query relevant, and factually aligned knowledge base, thereby increasing robustness and factual faithfulness.

CAF enhances SCF by providing fine-grained semantic validation. While SCF removes broad outliers based on statistical or semantic graph anomalies, CAF ensures that documents generated have coherent intent, correct facts, and logical consistency. This layered design increases the final output's robustness as well as its factual accuracy.

# 5 Experiments

## 5.1 Setup

This section describes the experimental setup. We evaluate the effectiveness and robustness of SeConRAG under various adversarial scenarios. All reported results are averages of multiple runs with an error of ± 1%.

**Datasets.** We test three popular open-domain question-answering benchmarks: Natural Questions (NQ) [24], HotpotQA [47], and MS-MARCO [6]. Each dataset corresponds to a large-scale corpus.

**Attack Settings.** To test robustness under poisoning scenarios, we evaluate two representative types of attacks against RAG systems: (1) Corpus Poisoning Attack, following PoisonedRAG [55], which inserts adversarial passages into the corpus; (2) Prompt Injection Attack (PIA) [53, 17],which adversarial prompts are created by perturbing discrete tokens to closely resemble training queries, thereby misleading the model during inference. It ensures a comprehensive evaluation of SeCon-RAG under both input and retrieval adversarial threats.

**Evaluation Metrics.** We use standard metrics from previous research to evaluate model robustness and answer quality: (1) Accuracy (ACC) is the percentage of generated answers that exactly match the ground truth. (2) Attack Success Rate (ASR): The percentage of poisoning queries or documents that cause the model to produce incorrect results.

**Verified Correct Documents.** To create the semantic reference set $D_{cor}$, we manually selected 10 clean documents from each dataset.

**Model.** We tested five LLMs from both open and closed source families: Mistral-12B, Qwen-7B, LLaMA-3.1-8B, GPT-4o, and DeepSeek-R1. RAG backbones are maintained in accordance with the corresponding LLMs. Appendix A.1 provides detailed prompts for EIRE, semantic similarity, and CAF modules. All experiments are carried out using NVIDIA A100-SXM4-40GB GPUs.

Table 1: Performance comparison of SeConRAG and baseline methods across three QA datasets and five LLMs under PIA, 20% and 100% corpus poisoning, and clean settings. Best values (highest accuracy ↑ or lowest ASR ↓) are highlighted in bold.

| Model | Method | HotpotQA [47] | | | | NQ [24] | | | | MS-MARCO [6] | | | |
|---|---|---|---|---|---|---|---|---|---|---|---|---|---|
| | | PIA ACC/ASR | PR-100% ACC/ASR | P-20% ACC/ASR | Clean ACC | PIA ACC/ASR | PR-100% ACC/ASR | PR20% ACC/ASR | Clean ACC | PIA ACC/ASR | PR-100% ACC/ASR | PR20% ACC/ASR | Clean ACC |
| Mistral-12B [4] | VanillaRAG | 51.0/40.0 | 0.9/98.2 | 38.2/58.0 | 75.0 | 47.6/37.5 | 8.2/90.9 | 38.2/48.2 | 68.0 | 54.5/43.6 | 9.1/89.1 | 50.0/45.5 | 84.0 |
| | InstructRAG [42] | 50.0/43.6 | 13.6/83.5 | 45.5/49.1 | 75.0 | 48.2/43.2 | 13.6/82.7 | 51.8/40.0 | 66.0 | 64.5/33.2 | 15.5/78.2 | 57.3/36.4 | 81.0 |
| | ASTUTERAG [39] | 68.2/17.3 | 32.7/61.1 | 65.9/21.8 | 76.0 | 64.5/10.0 | 43.6/38.2 | 67.7/11.8 | 70.0 | 75.9/17.3 | 32.7/58.2 | 73.6/18.8 | 81.0 |
| | TrustRAG [54] | 75.5/1.4 | 75.5/**3.6** | 71.8/14.5 | 81.0 | 68.2/**0.5** | 62.7/**1.8** | 66.4/13.6 | 73.0 | 90.9/**0.0** | **91.8**/**0.0** | 87.3/11.8 | 85.0 |
| | SeConRAG(ours) | **77.5/0.8** | **75.7**/**3.6** | **72.7/4.5** | **83.0** | **72.3**/1.8 | **63.6**/2.5 | **74.5/10.2** | **82.0** | **91.8/0.0** | 88.2/**0.0** | **89.1/9.1** | **98.0** |
| Qwen-7B [20] | VanillaRAG | 34.0/60.9 | 1.8/98.2 | 32.7/65.5 | 67.0 | 28.2/67.3 | 5.5/93.6 | 39.1/51.8 | 56.0 | 36.4/60.9 | 10.0/87.3 | 43.6/46.4 | 75.0 |
| | InstructRAG [42] | 58.2/38.2 | 24.5/76.4 | 45.5/51.8 | 67.0 | 52.7/45.5 | 25.5/76.4 | 47.3/47.3 | 64.0 | 61.8/36.4 | 43.6/57.8 | 49.1/45.5 | 75.0 |
| | ASTUTERAG [39] | 51.8/29.1 | 45.5/44.1 | 58.6/25.4 | 65.0 | 56.4/17.3 | 42.3/53.2 | 60.5/**17.3** | 68.0 | 44.5/45.5 | 42.3/54.5 | 65.5/20.0 | 74.0 |
| | TrustRAG [54] | 62.7/0.6 | 58.2/2.7 | 58.2/26.4 | 73.0 | 67.3/**0.6** | 60.0/2.7 | 64.5/24.5 | 67.0 | 75.5/1.4 | 71.8/4.5 | 75.5/17.5 | 78.0 |
| | SeConRAG(ours) | **67.3/0.5** | **63.6/2.3** | **61.8/21.8** | **76.0** | **73.6**/8.2 | **66.4/2.4** | **70.9**/21.8 | **78.0** | **75.5/1.4** | **71.8/4.5** | **75.5/17.5** | **84.0** |
| LLaMA-3.1-8B [13] | VanillaRAG | 31.8/62.7 | 4.5/96.4 | 36.4/57.3 | 70.0 | 38.2/54.5 | 10.9/88.2 | 41.8/52.7 | 70.0 | 34.5/63.6 | 9.1/88.2 | 54.5/40.9 | 83.0 |
| | InstructRAG [42] | 61.8/30.0 | 27.3/71.8 | 47.3/50.0 | 76.0 | 67.3/24.1 | 32.7/67.3 | 56.4/34.5 | 70.0 | 68.2/26.4 | 48.5/51.8 | 72.7/27.3 | 81.0 |
| | ASTUTERAG [39] | 43.6/41.8 | 46.8/47.0 | 65.5/20.9 | 68.0 | 57.3/26.4 | 58.2/31.8 | 77.5/8.2 | 81.0 | 59.1/39.5 | 56.8/38.6 | 82.3/13.6 | 89.0 |
| | TrustRAG [54] | 72.7/**0.5** | 67.3/**3.0** | 65.5/19.1 | 72.0 | 84.5/**0.2** | 79.1/**0.0** | 79.1/10.9 | 84.0 | 86.4/1.5 | 84.5/6.4 | 85.4/9.1 | 84.0 |
| | SeConRAG(ours) | **73.6/0.5** | **72.0**/10.9 | **67.4/18.4** | **84.0** | **85.1**/2.7 | **88.2**/1.8 | **86.9/4.0** | **90.0** | **87.3/0.2** | **89.1/0.0** | **86.2/9.1** | **90.0** |
| GPT-4o [1] | VanillaRAG | 57.3/40.0 | 11.9/81.8 | 45.5/30.5 | 81.0 | 50.9/44.3 | 27.3/68.2 | 52.7/31.8 | 74.0 | 70.0/27.3 | 30.0/64.1 | 72.3/16.4 | 84.0 |
| | InstructRAG [42] | 59.1/37.3 | 27.3/71.8 | 61.8/33.2 | 84.0 | 58.2/26.5 | 43.6/51.1 | 66.4/25.5 | 74.0 | 77.3/16.4 | 50.5/42.7 | 70.9/17.3 | 83.0 |
| | ASTUTERAG [39] | 72.7/14.5 | 67.3/24.1 | 77.3/11.8 | 81.0 | 83.6/4.5 | 75.5/14.2 | 79.1/4.1 | 81.0 | 90.5/0.7 | 76.4/15.5 | 82.7/6.4 | 86.0 |
| | TrustRAG [54] | 81.8/**0.3** | 80.9/2.7 | **79.1**/6.4 | 85.0 | 82.7/**0.3** | 80.0/**0.1** | 81.8/**1.0** | 86.0 | 89.1/1.3 | **89.1**/1.8 | 84.5/6.4 | 88.0 |
| | SeConRAG(ours) | **83.6/0.3** | **83.6/2.4** | **79.1/5.5** | **86.0** | **89.1**/0.6 | **81.8/0.0** | **84.5/1.0** | **88.0** | **93.6/0.0** | **89.1/1.8** | **89.1/3.6** | **94.0** |
| DeepSeek-R1 [18] | VanillaRAG | 59.1/32.7 | 10.0/89.1 | 51.0/46.4 | 81.0 | 64.3/27.3 | 17.3/84.5 | 51.0/43.6 | 80.0 | 71.8/25.5 | 11.8/81.8 | 60.5/29.1 | 82.0 |
| | InstructRAG [42] | 61.8/34.5 | 27.3/72.7 | 61.8/38.2 | 80.0 | 59.1/28.2 | 39.1/62.7 | 65.5/32.7 | 82.0 | 75.5/18.2 | 51.8/47.5 | 72.7/26.4 | 87.0 |
| | ASTUTERAG [39] | 73.6/14.5 | 64.5/25.5 | 77.3/14.5 | 79.0 | 90.0/1.8 | 81.8/10.9 | 89.1/**0.0** | 87.0 | 80.3/4.5 | 85.5/8.2 | 89.1/5.5 | 88.0 |
| | TrustRAG [54] | 81.8/4.5 | 79.1/**2.7** | **85.5**/10.0 | **89.0** | 90.0/1.8 | 88.2/**0.0** | 90.0/3.6 | 91.0 | **93.6**/1.8 | 89.1/3.6 | 89.1/5.5 | 91.0 |
| | SeConRAG(ours) | **84.5**/3.0 | **81.8**/8.0 | 83.6/**5.5** | 86.0 | **90.0/0.9** | **96.4/0.0** | **96.4/0.0** | **98.0** | 92.7/3.0 | **94.5/1.8** | **94.5/5.5** | **94.0** |

## 5.2 Main Results

We evaluate the robustness and effectiveness of SeconRAG against four representative Retrieval-Augmented Generation (RAG) defense baselines VanillaRAG, InstructRAG [42], AstuteRAG [39], and TrustRAG [54], across three datasets and five LLMs. To verify that our method can handle both high and low poisoning rates simultaneously, each model is tested under four settings: Clean, Prompt Injection Attack, PoisonedRAG-20%, and PoisonedRAG-100%, with results reported in terms of Accuracy and Attack Success Rate. Table 1 summarizes the key findings. More detailed PoisonedRAG results at different poisoning levels can be found in Appendix A.4.1.

Results in Table 1 demonstrates that SeConRAG outperforms in almost all datasets, LLMs, and attack scenarios. Under high poisoning (PoisonedRAG-100%), it consistently maintains high accuracy and low ASR. For example, on HotpotQA with GPT-4o, SeConRAG achieves 83.6% accuracy and 2.4% ASR, outperforming TrustRAG (80.9% / 2.7%) and ASTUTERAG (67.3% / 24.1%). A Similar trends hold under low poisoning (20%), where SeConRAG consistently improves robustness compared with baselines. SeConRAG also performs well against Prompt Injection Attacks, which target the input layer rather than retrieval. On MS-MARCO with GPT-4o, it achieves 93.6% accuracy and 0.0% ASR, slightly surpassing TrustRAG and ASTUTERAG. Even with smaller models such as Qwen-7B, SeConRAG retains competitive performance (67.3% / 0.5%), demonstrating the effectiveness of CAF in mitigating prompt-level inconsistency.

Importantly, SeConRAG continues to perform well on clean corpora. On MS-MARCO with DeepSeek-R1, it reaches 94.0% accuracy, and on NQ with Mistral-12B, it achieves 82.0%, outperforming TrustRAG (73.0%) and ASTUTERAG (70.0%). This demonstrates that the defense mechanisms do not degrade benign performance. Overall, SeConRAG consistently outperforms or equals existing defenses across datasets, LLMs, and threat scenarios. Across both large models (GPT-4o, DeepSeek-R1) and smaller instruction-tuned models (Qwen-7B, Mistral-12B), it consistently delivers reliable and generalizable robustness against both corpus poisoning and prompt-level adversaries, making it a viable defense for real-world RAG deployments.

## 5.3 Ablation Study

We conduct an ablation study on the Mistral-12B model to evaluate the contributions of SeCon-RAG's components.We concentrate on two key areas: (i) the core SCF and CAF modules, and (ii) an ablation study that includes the SCF subcomponents, EIRE, and the verified Correct Document Set.

### 5.3.1 Core Modules (SCF and CAF).

To evaluate the impact of SCF, we remove this module and compare performance with three QA datasets. SCF uses clustering and semantic graph filtering to eliminate documents that are semantically irrelevant or poisoning. Figure 3 shows that disabling SCF consistently decreases accuracy and

increases ASR across all datasets. In the 100% poisoning setting on HotpotQA, accuracy drops from 74.0% to 71.0%, and ASR increases from 8.0% to 25.0%. Under prompt injection attacks, accuracy decreases from 92.0% to 85.0%. These findings support SCF's effectiveness in increasing retrieval precision and resisting semantically attacks. We then assess the CAF module, which filters semantically conflicting evidence using EIRE-based consistency checks. Removing CAF leads to more severe degradation. When using HotpotQA with 100% poisoned data, accuracy drops to 68.0% and ASR rises to 56.0%. ASR increases to 47.0% on NQ, while accuracy decreases from 92.0% to 46.0% on PIA. These findings highlight CAF's critical role in detecting and filtering conflicting or misleading documents that SCF alone may miss.

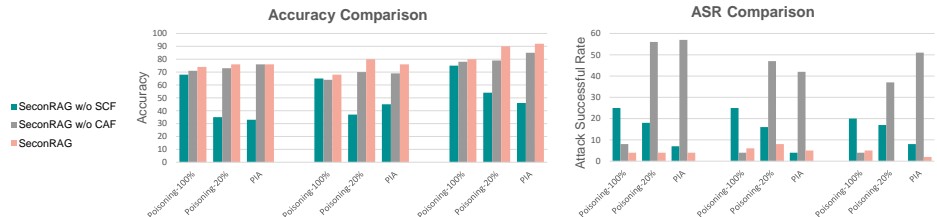

Figure 3: Ablation results on accuracy and attack success rate (ASR) across three datasets using Mistral-12B. From left to right are HotpotQA, NQ, MS-MARCO.

### 5.3.2 Evaluation of SCF Subcomponents, EIRE, and the Verified Correct Document Set.

To examine whether clustering and semantic filtering are complementary, we test each individually. As shown in Table A.4.1, their combination yields the strongest robustness, achieving lower ASR in several cases, which confirms the necessity of combination.

In addition, we assess the standalone effectiveness of the Entity-Intent-Relation Extractor and the verified correct document set ($d_{cor}$). Table A.4.2 in appendix summarizes the results. EIRE improves the fine-grained reasoning capability of both SCF and CAF. With EIRE enabled, the model consistently achieves higher factual accuracy while significantly lowering ASR, especially under high poisoning conditions. Similarly, a small, high-quality $d_{cor}$ set can significantly improve semantic filtering performance and reduce noise from poisoned documents, as well as improve robustness under high-poisoning conditions (e.g., ASR $\rightarrow$ 0 on MS-MARCO 100% poisoning).

The ablation results show that both SCF and CAF are critical for protecting against poisoning attacks. SCF performs coarse filtering of anomalous content, while CAF ensures semantic and factual consistency. Their collaboration allows SeCon-RAG to maintain strong performance in high-poisoning and adversarial scenarios.

### 5.4 Runtime Analysis

We compare SeConRAG's runtime cost to four representative RAG baselines—VanillaRAG, InstructRAG, AstuteRAG, and TrustRAG—on three QA benchmarks: HotpotQA, NQ, and MS-MARCO. The methods are evaluated in three adversarial settings: Prompt Injection Attack and PoisonedRAG with 100% or 20% poisoning. Figure 4 depicts the full results. Although SeConRAG achieves

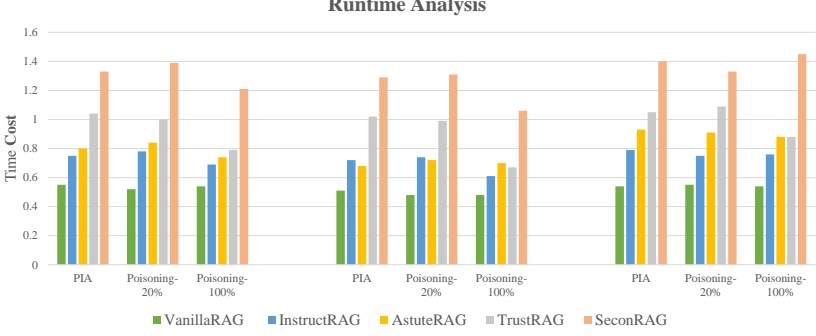

Figure 4: The average runtime per batch (in minutes) for three datasets and adversarial settings. From left to right are HotpotQA, NQ, MS-MARCO.

the highest robustness in all settings, it has a moderate runtime overhead. It takes between 1.21 and 1.45 minutes per batch, depending on the dataset and the severity of the attack. This cost is due to its multi-stage semantic filtering, consistency checks, and conflict filtering, which protects against poisoned documents. Despite the additional cost, SeConRAG maintains a practical runtime range. Deeper semantic understanding requires the use of LLMs for semantic structure extraction and graph similarity calculation. Despite the additional cost, SeConRAG has a reasonable runtime range. This trade-off is acceptable for many real-world RAG applications that need both robustness and correctness. For example, on NQ with 100% poisoning, it completes in 1.06 minutes, only 10 seconds slower than TrustRAG (0.67 min) or AstuteRAG (0.70 min), but it offers significantly more reliable answers. Overall, the asymptotic overhead is moderate relative to standard retrieval. This trade-off is acceptable for many real-world RAG applications that require robustness and correctness.

## 5.5 Embedding Models

We further evaluate SeConRAG with four widely used embedding models: MiniLM[41], SimCSE[16], BERT[11], and BGE[8]. These encoders are integrated into the retrieval and two-stage filtering pipelines, with Mistral-12B serving as the primary LLM. Table 2 displays results from three datasets with various poisoning ratios. Across different embedding model, SeConRAG maintains high accuracy ($> 75\%$) and low ASR ($< 10\%$) under 100% poisoning. For example, on MS-MARCO, BGE achieves 90.0%/0.0%, while MiniLM yields 77.3%/7.3%. These findings confirm that SeConRAG's defense framework perform well across embeddings and avoids reliance on a single model.

Table 2: Comparison of SeConRAG performance under different embedding models (MiniLM, SimCSE, BERT, BGE) across varying poisoning ratios on three datasets. highest accuracy ↑ or lowest ASR ↓

| Model | Setting | HotpotQA [47] | | | | | NQ [24] | | | | | MS-MARCO [6] | | | | |
|---|---|---|---|---|---|---|---|---|---|---|---|---|---|---|---|---|
| | | 100% | 80% | 60% | 40% | 20% | 100% | 80% | 60% | 40% | 20% | 100% | 80% | 60% | 40% | 20% |
| | | ACC/ASR | ACC/ASR | ACC/ASR | ACC/ASR | ACC/ASR | ACC/ASR | ACC/ASR | ACC/ASR | ACC/ASR | ACC/ASR | ACC/ASR | ACC/ASR | ACC/ASR | ACC/ASR | ACC/ASR |
| mistral-12b | SimCSE | 73.6 / 8.2 | 77.3 / 4.0 | 75.5 / 4.0 | 71.8 / 8.2 | 73.6 / 4.0 | 67.3 / 5.5 | 67.3 / 0.0 | 67.3 / 3.6 | 69.1 / 0.0 | 79.1 / 7.3 | 79.1 / 7.3 | 91.8 / 1.8 | 91.8 / 0.0 | 90.0 / 1.8 | 90.0 / 0.0 |
| mistral-12b | MiniLM | 75.5 / 9.1 | 75.5 / 5.5 | 77.3 / 5.5 | 77.3 / 5.5 | 75.5 / 4.0 | 75.5 / 3.6 | 71.8 / 5.5 | 71.8 / 0.0 | 69.1 / 1.8 | 70.9 / 0.0 | 77.3 / 7.3 | 91.8 / 1.8 | 90.0 / 0.0 | 90.0 / 0.0 | 91.8 / 0.0 |
| mistral-12b | BGE | 75.5 / 5.5 | 77.3 / 4.0 | 75.5 / 4.0 | 75.5 / 7.3 | 71.8 / 9.1 | 70.9 / 9.1 | 67.3 / 1.8 | 71.8 / 0.0 | 71.8 / 1.8 | 73.6 / 0.0 | 90.0 / 0.0 | 91.8 / 0.0 | 90.0 / 0.0 | 91.8 / 0.0 | 90.0 / 9.1 |
| mistral-12b | BERT | 72.7 / 6.4 | 77.3 / 7.3 | 75.5 / 4.0 | 75.5 / 5.5 | 75.5 / 11.5 | 74.5 / 10.9 | 67.3 / 7.3 | 71.8 / 3.6 | 69.1 / 1.8 | 69.1 / 1.8 | 79.1 / 9.1 | 89.1 / 1.8 | 91.8 / 0.0 | 91.8 / 0.0 | 93.6 / 0.0 |

## 5.6 Sensitivity Analysis of Filtering Thresholds

We investigate the impact of two primary thresholds, $\tau_{cluster}$ and $\tau_{semantic}$, on LLaMA-3.1-8B and GPT-4o. As shown in Table A.4.4, performance remains stable across reasonable ranges ($\tau_{cluster} \in [0.86, 0.90]$, $\tau_{semantic} \in [0.2, 0.4]$), with accuracy variations within $\pm 2\%$ and low ASR. This robustness is due to the conservative AND-logic in joint filtering, which ensures that only documents flagged by both filters are removed. As a result, SeConRAG is not overly sensitive to precise hyperparameter tuning, making it useful in real-world deployment.

# 6 Conclusion

We propose SeCon-RAG, a robust retrieval-augmented generation framework that protects against corpus poisoning. It combines two complementary modules: Semantic and Cluster-Based Filtering, which removes poisoned content using clustering and semantic similarity, and Conflict-Aware Filtering, which filters out contradictory or misleading evidence using structured semantic reasoning. Experiments with multiple datasets and poisoning scenarios show significant improvements in answer accuracy and reduced attack success rates. SeCon-RAG provides a scalable and interpretable defense for RAG systems in adversarial environments by combining coarse-grained statistical pruning and fine-grained semantic validation. The Impact Statement of our paper is shown in the appendix.

**Limitations.** While SeCon-RAG demonstrates strong robustness against a range of poisoning attacks, several limitations remain. First, SeCon-RAG introduces moderate inference latency due to multiple LLM calls (EIRE extraction, semantic similarity, and CAF decision-making). Second, the framework relies on high-quality semantic extraction; Finally, a small set of manually verified documents $D_{cor}$ is required.Future research could reduce runtime overhead by replacing EIRE with smaller models and exploring lightweight graph similarity metrics. These changes will make SeCon-RAG better suited for latency-sensitive, real-time RAG applications.

## Acknowledgments and Disclosure of Funding

This work is supported by CAS Project for Young Scientists in Basic Research, Grant No.YSBR-040, ISCAS New Cultivation Project ISCAS-PYFX-202201, ISCAS Basic Research ISCAS-JCZD-202302, National Natural Science Foundation of China General Project (No. 71971051, No. 72371067),the National Research Foundation, Singapore, and DSO National Laboratories under the AI Singapore Programme (AISG Award No: AISG4-GC-2023-008-1B); by the National Research Foundation Singapore and the Cyber Security Agency under the National Cybersecurity R&D Programme (NCRP25-P04-TAICeN); and by the Prime Minister's Office, Singapore under the Campus for Research Excellence and Technological Enterprise (CREATE) Programme. Any opinions, findings and conclusions, or recommendations expressed in these materials are those of the author(s) and do not reflect the views of the National Research Foundation, Singapore, Cyber Security Agency of Singapore, Singapore.

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

# A   Appendix / supplemental material

**Impact Statement**

SeCon-RAG's effectiveness is dependent on the quality of its semantic parser (EIRE), which may perform poorly on domain-specific texts. The methods proposed in this paper will not have a negative impact on the community.

## A.1   Prompt and Example

### A.1.1   EIRE (Entity-Intent-Relation Extractor)

We provide the prompt template used in our system to extract the intent, key entities, and entity relations from a given documents.

---

**Prompt for EIRE**

Please extract both the key entities, intent and relations of entities of the documents, using the following criteria:

**Key Entities**: Extract specific entities (such as terms, names, data, and locations) that are explicitly mentioned in the document for later entity matching.
**Intent**: Summarize the document's main points or conclusion in a single sentence. This should be free of external issues and only include the document's own claims.
**Relation**: Identify the most critical relationship between key entities in the document, keeping the length under the original document.

---

**Output for example document extracted by EIRE**

**Input:**   L'Oiseau Blanc (commonly known in the English-speaking world as The White Bird ) was a French Levasseur PL.8 biplane that disappeared in 1927, during an attempt to make the first non-stop transatlantic flight between Paris and New York City to compete for the Orteig Prize. The aircraft was flown by French World War I aviation heroes, Charles Nungesser and FranŎ0e7ois Coli. The aircraft took off from Paris on 8 May 1927 and was last seen over Ireland. Less than two weeks later, Charles Lindbergh successfully made the New YorkŽ2013Paris journey and claimed the prize, flying the Špirit of St. Louis
**Output:**

**Intent:**
```
"L'Oiseau Blanc disappeared in 1927 while attempting the first non-stop
transatlantic flight from Paris to New York to win the Orteig Prize."
```
**Entities:**
```
"L'Oiseau Blanc", "Levasseur PL.8", "1927", "Charles Nungesser", "Paris"
"New York City", "Orteig Prize", "non-stop transatlantic flight",
"Ireland", "Charles Lindbergh", "François Coli",  "Spirit of St. Louis"
```
**Relations:**
```
    ["L'Oiseau Blanc", "Levasseur PL.8", "is_a"],
    ["L'Oiseau Blanc", "1927", "disappeared_in"],
    ["L'Oiseau Blanc", "non-stop transatlantic flight", "attempt"],
    ["L'Oiseau Blanc", "Paris", "departed_from"],
    ["L'Oiseau Blanc", "New York City", "destination"],
    ["L'Oiseau Blanc", "Charles Nungesser", "flown_by"],
    ["L'Oiseau Blanc", "François Coli", "flown_by"],
    ["L'Oiseau Blanc", "Ireland", "last_seen_over"],
    ["Charles Lindbergh", "Orteig Prize", "won"],
    ["Charles Lindbergh", "Spirit of St. Louis", "flew"]
```

---

### A.1.2 Semantic filtering

Evaluate if the following document contains deceptive or malicious material that may impact the answer to the inquiry. Consider the presence of certain terms, their relationships, and any underlying meanings.

**Use the graph generated using these semantic information and to calculate its the structure's similarity between the correct document which is reference**

```
reference documents:
"intent":"L'Oiseau Blanc was a French biplane that disappeared in 1927"

"keywords":
["L'Oiseau Blanc", "Levasseur PL.8", "1927", "Charles Nungesser",
"non-stop transatlantic flight", "Paris",
"New York City", "Orteig Prize",  "Spirit of St. Louis"
"François Coli", "Ireland", "Charles Lindbergh"]

"relations":
[("L'Oiseau Blanc", "Levasseur PL.8", "is_a"),
("L'Oiseau Blanc", "1927", "disappeared_in"),
("L'Oiseau Blanc", "non-stop transatlantic flight", "attempt"),
("L'Oiseau Blanc", "Paris", "departed_from"),
("L'Oiseau Blanc", "New York City", "destination"),
 ("L'Oiseau Blanc", "Charles Nungesser", "flown_by"),
("L'Oiseau Blanc", "François Coli", "flown_by"),
("L'Oiseau Blanc", "Ireland", "last_seen_over"),
("Charles Lindbergh", "Orteig Prize", "won"),
("Charles Lindbergh", "Spirit of St. Louis", "flew") ]
```

Score and filter documents based on their similarities and subsequent evaluations. Give a score between 0 and 10, where 0 indicates absolutely benign and 10 implies highly misleading or malevolent. The output must only show your score as an integer number.
Analysis Criteria:
Identify entity/Phrases: Identify entity or phrases that could be misleading or indicate a departure from reality.
Relationship to the Question: Analyze how these terms relate to the question and determine if they might affect the answer to the question.
Hidden Meanings or Implications: Consider if the document suggests fictional or inaccurate information without clear indication. Context: Consider how the document might be interpreted in the context of the question. Assess whether it could mislead someone seeking factual information.
Intent and Impact: Evaluate if the misleading information appears intentional and whether it could cause harm or spread misinformation.
Ambiguity and Clarity: Assess the document's clarity. Vague or misleading presentations can be more harmful.
Fiction vs. Reality: Determine if the document clearly distinguishes between hypothetical scenarios and reality.
Relevance to the Question: Evaluate if the misleading information directly impacts the question's answer.
Scoring Guidelines: 0-3: Likely benign. The document contains little to no misleading or harmful information related to the question. 4-6: Potentially misleading. The document contains some elements that may confuse or mislead, but they are not the primary focus or are clearly indicated as hypothetical or fictional. 7-10: Likely malicious or misleading. The document actively presents information that could mislead, deceive, or significantly impact the answer to the question. Considerations for Scoring:
Final Score: Provide your score based on the above analysis, Provide your score as an integer number only.

The distribution of the semantic graphs produced from semantic information in vector space is depicted in the image below. We utilize PCA to reduce the vector's dimension to two dimensions and demonstrate it. Figure 5, 6, 7 visualize semantic graphs generated by EIRE for correct and poisoned documents under the query: *"Which French ace pilot and adventurer flew L'Oiseau Blanc?"* .

We employ the prompt of Semantic graph for filtering A.1.2 to direct the llms in evaluating, scoring and filtering documents based on semantic information and correspoding graphs.

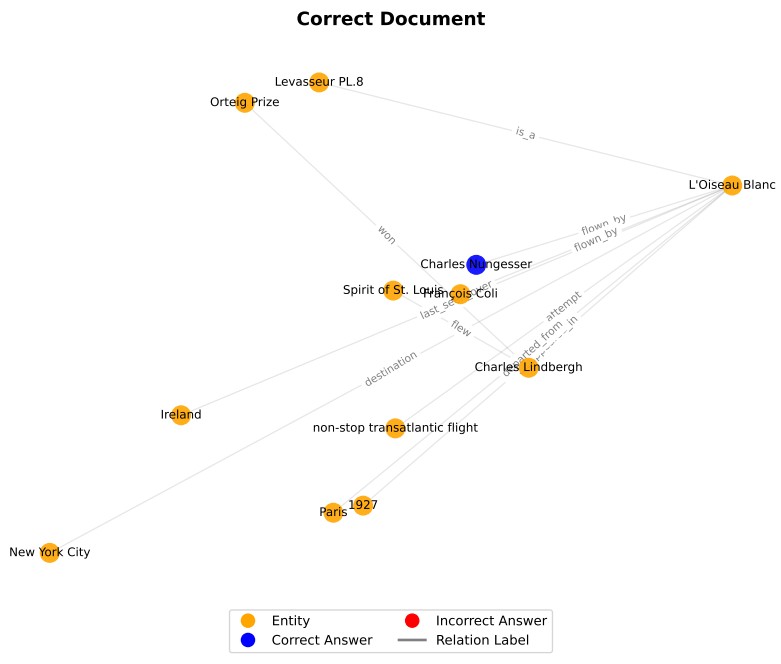

Figure 5: A schematic graph of the correct document's semantic structure in two dimensions. Blue indicates the correct response, orange nodes stand for entities, and the edges connecting nodes show the connections between entities.

---

**Example for correct document**

**Input:**

```
"Can't Help Falling in Love\" is a pop ballad originally recorded by
American singer Elvis Presley and published by Gladys Music,
Presley's publishing company. It was written by Hugo Peretti,
Luigi Creatore, and George David Weiss.[2] The melody is based
on \"Plaisir d'amour\",[4] a popular romance by Jean-Paul-\u00c9gide
Martini (1784). The song was featured in Presley's 1961 film,
Blue Hawaii. During the following four decades,
it was recorded by numerous other artists, including Tom Smothers,
Swedish pop group A-Teens, and the British reggae group UB40,
whose 1993 version topped the U.S. and UK charts."
```

**Output:**

```
Semantic Content: Factually accurate, no false claims.
Graph Similarity: Completely unrelated (topic is music, not aviation).
Misleading Risk: Low although off-topic, it doesn't mislead  facts.
Final Score: 2
```

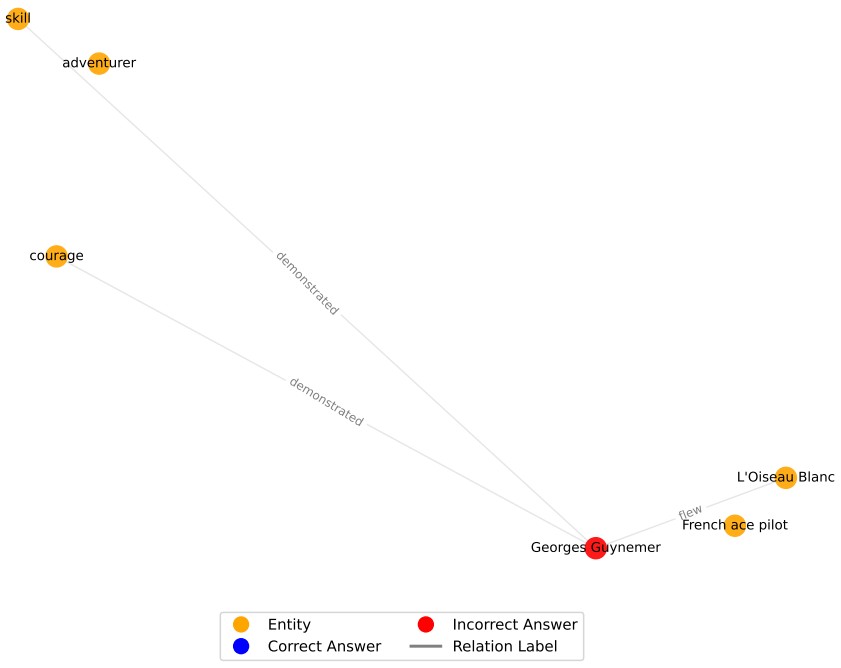

Figure 6: A schematic graph of the incorrect document's semantic structure in two dimensions. Red indicates the poisoning response, orange nodes stand for entities, and the edges connecting nodes show the connections between entities.

---

**Example for correct document**

**Input:**
```
"Frank Sinatra, the iconic crooner, recorded an unforgettable
rendition of the song \"I Can't Help Falling in Love With You\",
enrapturing audiences with his soulful interpretation."
```
**Output:**
```
Semantic Content: Highly confident and emotional false claim.
Graph Similarity: No overlap with reference.
Misleading Risk:High persuasive wording increases belief in a falsehood.
Final Score: 8
```

## A.2 Conflict-Aware Filtering

In the final inference process, we prompt the LLMs to determine which information from the retrieve documents is reliable from three dimensions using the semantic information extracted by EIRE.

> **Output for example document extracted by EIRE**
>
> You must evaluate the document information you retrieve, which includes internal knowledge, external knowledge, and query.
>
> To answer this question, consider intent, key entities, and relationships to determine which knowledge provides the best, most accurate, and error-free support.
>
> The external information may not be reliable. Use a combination of intent and key entities from external information, as well as the intent of the original question, to make self-judgments about the reliability of external information. Then, based on both your assessment and your personal knowledge, provide the best possible answer.

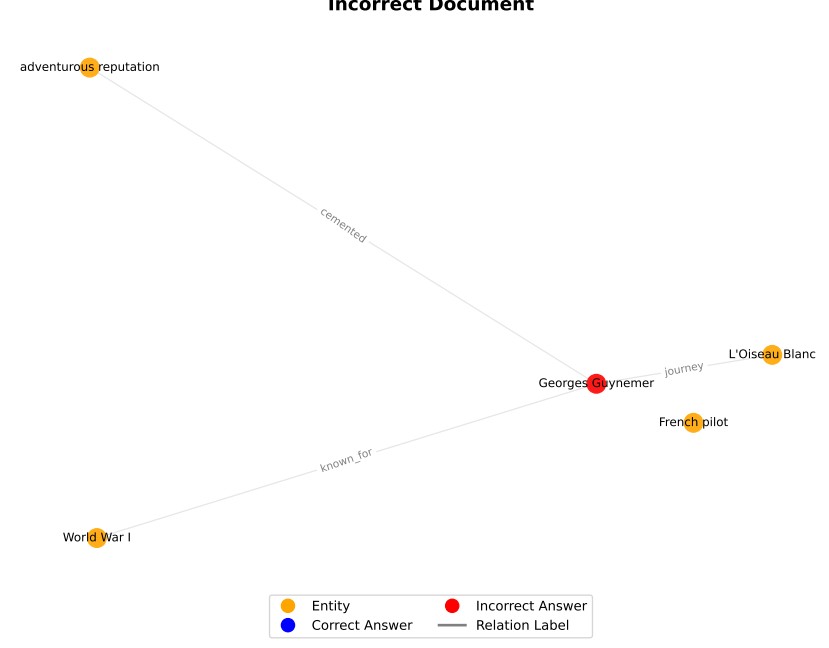

Figure 7: A schematic graph of the incorrect document's semantic structure in two dimensions. Red indicates the poisoning response, orange nodes stand for entities, and the edges connecting nodes show the connections between entities.

### A.3 Pseudocode of SeCon-RAG

Provide formally written pseudocode (see Algorithm 1) for the full SeCon-RAG pipeline, including SCF and CAF. This helps clarify the implementation logic for reproducibility.

### A.4 Experiments

#### A.4.1 Experiments of Different Poisoning Ratio

HotpotQA Table 3 compares SeConRAG's performance to four baseline methods (VanillaRAG, InstructRAG, ASTUTERAG, and TrustRAG) across five backbone LLMs on the HotpotQA dataset with varying corpus poisoning ratios (0% to 100%). Across all models and poisoning levels, SeConRAG consistently achieves or approaches the highest accuracy while maintaining low attack success rates (ASR), demonstrating strong robustness and generalizability. Notably, On Mistral-12B SeConRAG achieves 75.7% accuracy with only 3.6% ASR under 100% poisoning, outperforming TrustRAG and significantly surpassing ASTUTERAG and InstructRAG. On GPT-4o, SeConRAG achieves the highest accuracy (83.6%) and lowest ASR (2.4%) under full poisoning, indicating its effectiveness

**Algorithm 1** SeCon-RAG: Two-Stage Semantic Filtering and Conflict-Aware Generation

---

**Require:** Query $q$, Retrieval corpus $\mathcal{D}$, Verified clean documents $\mathcal{D}_{\text{cor}}$, Pretrained LLM of RAG $F$
**Ensure:** Trustworthy answer $A(q)$

    **Stage 1: Semantic and Cluster-Based Filtering (SCF)**

1: Embed each document $d \in \mathcal{D}$ into vector $m(d)$
2: Apply K-Means clustering to obtain clusters $\mathcal{C} = \{c_1, \ldots, c_K\}$
3: **for all** $d \in \mathcal{D}$ **do**
4:     Compute similarity to cluster centroid: $s_{\text{cluster}}(d) \leftarrow \text{sim}(m(d), \mu_c)$
5:     Extract semantic structure $(E_d, I_d, R_d) \leftarrow \text{EIRE}(d)$
6:     Construct semantic graph $G_d$ from $(E_d, I_d, R_d)$
7:     Compute semantic similarity score $s_{\text{sem}}(d) \leftarrow \text{LLM}(G_d, \mathcal{G}_{\text{cor}})$
8: **end for**
9: Filter documents where $s_{\text{cluster}}(d) > \tau_{\text{cluster}}$ **and** $s_{\text{sem}}(d) < \tau_{\text{sem}}$
10: Define filtered corpus $\widetilde{\mathcal{D}} \leftarrow \mathcal{D} \setminus \mathcal{D}_{\text{filtered}}$

    **Stage 2: Conflict-Aware Filtering (CAF)**

11: Retrieve top-$k$ documents $\mathcal{D}_k(q)$ from $\widetilde{\mathcal{D}}$ based on embedding similarity
12: **for all** $d \in \mathcal{D}_k(q)$ **do**
13:     Extract semantic structure $(E_d, I_d, R_d) \leftarrow \text{EIRE}(d)$
14:     Evaluate:

        • Query consistency $Q(d, q)$
        • Corpus consistency $C(d, \mathcal{D}_k(q))$
        • Model consistency $M(d, F)$

15:     **if** $\text{CAF}(d, Q, C, M) = \texttt{trustable}$ **then**
16:         Add $d$ to $\mathcal{D}_{\text{CAF}}$
17:     **end if**
18: **end for**
19: Generate final answer: $A(q) \leftarrow F(q, \mathcal{D}_{\text{CAF}})$
20: **return** $A(q)$

---

even with strong LLMs. On smaller models such as Qwen-7B and LLaMA-3.1-8B, SeConRAG maintains competitive performance, outperforming all baselines under medium and low poisoning, demonstrating its scalability across model sizes. Under clean settings (0% poisoning), SeConRAG performs well and achieves high accuracy, indicating that the two-stage filtering does not overly suppress useful content.

Natural Questions (NQ) Table 4 compares SeConRAG's performance to baseline methods across five language models on the Natural Questions (NQ) benchmark, with six poisoning levels ranging from 0% (clean) to 100% . Through all LLMs and poisoning levels, SeConRAG consistently outperforms baseline methods in terms of answer accuracy and attack robustness. On Mistral-12B, SeConRAG outperforms TrustRAG and ASTUTERAG in both metrics, achieving up to 82.0% accuracy on clean data and maintaining high performance under attack (74.5% at 20% poisoning with only 10.2% ASR). Even with a smaller model, SeConRAG shows significant improvement. It achieves 78.0% accuracy on clean data and is more robust to 100% poisoning (66.4% / 2.4%) than TrustRAG (60.0% / 2.7%) and ASTUTERAG (42.3% / 53.2%). SeConRAG achieves 90.0% accuracy on clean data and 90.0% under 60% poisoning with 0.0% ASR, outperforming all baselines at almost every poisoning level on LLaMA-3.1-8B. On GPT-4o or DeepSeek-R1, SeConRAG outperforms at low-to-medium poisoning levels while maintaining low ASR across all ratios. SeConRAG outperforms TrustRAG and ASTUTERAG by achieving 100.0% accuracy with 0.0% ASR at 40% poisoning and over 96% accuracy with 0.0% ASR under full (100%) poisoning. These findings demonstrate SeConRAG's ability to maintain high factual accuracy while resisting poisoning attacks. Its consistent performance in both clean and adversarial environments demonstrates the effectiveness of the two-stage SCF and CAF filtering mechanisms.

MS-MARCO Table 5 compares the performance of SeConRAG and baseline RAG defense methods on the MS-MARCO dataset at different corpus poisoning ratios (0% to 100%). SeConRAG consistently delivers the best or near-best performance in all settings. Mistral-12B: SeConRAG

Table 3: Performance comparison of SeConRAG and baseline methods on HotpotQA using different Poisoning RAG ratios (highest accuracy ↑ or lowest ASR ↓).

| Model | Method | HotpotQA [47] | | | | | |
|---|---|---|---|---|---|---|---|
| | | 100% (ACC↑ / ASR↓) | 80% (ACC↑ / ASR↓) | 60% (ACC↑ / ASR↓) | 40% (ACC↑ / ASR↓) | 20% (ACC↑ / ASR↓) | 0% (ACC↑) |
| Mistral-12B [4] | VanillaRAG | 0.9 / 98.2 | 9.1 / 90.0 | 11.8 / 86.4 | 21.8 / 74.5 | 38.2 / 58.0 | 75.0 |
| | InstructRAG [42] | 13.6 / 83.5 | 23.6 / 71.8 | 25.5 / 70.0 | 37.3 / 57.3 | 45.5 / 49.1 | 75.0 |
| | ASTUTERAG [39] | 32.7 / 61.1 | 40.0 / 55.5 | 47.3 / 50.0 | 55.5 / 35.5 | 65.9 / 21.8 | 76.0 |
| | TrustRAG [54] | 75.5 / **3.6** | 74.5 / 5.5 | 78.2 / 4.5 | **74.5 / 6.4** | 71.8 / 14.5 | 81.0 |
| | SeconRAG(ours) | **75.7 / 3.6** | **77.3 / 4.5** | 75.5 / 4.5 | 71.8 / 8.2 | 72.7 / 4.5 | **83.0** |
| Qwen-7B [20] | VanillaRAG | 1.8 / 98.2 | 9.1 / 90.0 | 14.5 / 85.5 | 23.6 / 75.5 | 32.7 / 65.5 | 67.0 |
| | InstructRAG [42] | 24.5 / 76.4 | 30.9 / 69.1 | 31.8 / 68.2 | 35.5 / 63.6 | 45.5 / 51.8 | 67.0 |
| | ASTUTERAG [39] | 45.5 / 44.1 | 44.5 / 43.6 | 46.4 / 42.7 | 50.9 / 35.5 | 58.6 / 25.4 | 65.0 |
| | TrustRAG [54] | 58.2 / 2.7 | 64.5 / 4.5 | 69.1 / 4.5 | 65.5 / 3.6 | 58.2 / 26.4 | 73.0 |
| | SeconRAG(ours) | **63.6 / 2.3** | **67.3 / 1.8** | **73.6 / 3.6** | **67.3 / 2.7** | **61.8 / 21.8** | **76.0** |
| LLaMA-3.1-8B [13] | VanillaRAG | 4.5 / 96.4 | 25.5 / 74.5 | 30.0 / 68.2 | 42.7 / 63.6 | 36.4 / 57.3 | 70.0 |
| | InstructRAG [42] | 27.3 / 71.8 | 42.7 / 54.5 | 51.8 / 46.4 | 49.1 / 48.2 | 47.3 / 50.0 | 76.0 |
| | ASTUTERAG [39] | 46.8 / 47.0 | 52.7 / 40.0 | 53.6 / 38.2 | 62.7 / 29.1 | 65.5 / 20.9 | 68.0 |
| | TrustRAG [54] | 67.3 / **3.0** | 65.5 / 7.3 | 68.2 / 6.4 | 71.8 / 5.5 | 65.5 / 19.1 | 72.0 |
| | SeconRAG(ours) | **72.0** / 10.9 | **78.2 / 4.5** | **75.5 / 3.6** | **77.3 / 1.8** | 67.4 / 18.4 | **84.0** |
| GPT-4o [1] | VanillaRAG | 11.9 / 81.8 | 32.7 / 57.3 | 46.4 / 50.0 | 48.2 / 43.6 | 45.5 / 30.5 | 81.0 |
| | InstructRAG [42] | 27.3 / 71.8 | 46.4 / 50.0 | 48.2 / 49.1 | 55.5 / 40.9 | 61.8 / 33.2 | 84.0 |
| | ASTUTERAG [39] | 67.3 / 24.1 | 73.6 / 15.5 | 77.3 / 12.7 | 78.2 / 10.0 | 77.3 / 11.8 | 81.0 |
| | TrustRAG [54] | 80.9 / 2.7 | **83.6 / 3.6** | 81.8 / 3.6 | 81.8 / 3.6 | **79.1** / 6.4 | 85.0 |
| | SeconRAG(ours) | **83.6 / 2.4** | 82.7 / 4.5 | **83.6 / 4.5** | **83.6 / 1.8** | **79.1 / 5.5** | **86.0** |
| DeepSeek-R1 [18] | VanillaRAG | 10.0 / 89.1 | 31.8 / 67.3 | 35.5 / 61.8 | 40.9 / 55.5 | 51.0 / 46.4 | 81.0 |
| | InstructRAG [42] | 27.3 / 72.7 | 48.2 / 51.8 | 57.3 / 42.7 | 56.4 / 42.7 | 61.8 / 38.2 | 80.0 |
| | ASTUTERAG [39] | 64.5 / 25.5 | 66.4 / 24.5 | 72.7 / 18.2 | 72.7 / 17.3 | 77.3 / 14.5 | 79.0 |
| | TrustRAG [54] | 79.1 / **2.7** | 81.8 / 5.5 | 86.4 / **1.8** | **82.7** / 2.7 | **85.5** / 10.0 | **89.0** |
| | SeconRAG(ours) | **81.8** / 8.0 | **83.6 / 3.6** | **87.3** / 3.6 | 82.7 / 3.6 | 83.6 / **5.5** | 86.0 |

Table 4: Performance comparison of SeConRAG and baseline methods on NQ using different Poisoning RAG ratios (highest accuracy ↑ or lowest ASR ↓).

| Model | Method | NQ [24] | | | | | |
|---|---|---|---|---|---|---|---|
| | | 100% (ACC↑ / ASR↓) | 80% (ACC↑ / ASR↓) | 60% (ACC↑ / ASR↓) | 40% (ACC↑ / ASR↓) | 20% (ACC↑ / ASR↓) | 0% (ACC↑) |
| Mistral-12B [4] | VanillaRAG | 8.2 / 90.9 | 10.9 / 87.3 | 14.5 / 80.0 | 29.1 / 65.5 | 38.2 / 48.2 | 68.0 |
| | InstructRAG [42] | 13.6 / 82.7 | 17.3 / 78.2 | 26.4 / 70.0 | 38.2 / 56.4 | 51.8 / 40.0 | 66.0 |
| | ASTUTERAG [39] | 43.6 / 38.2 | 50.9 / 32.7 | 53.6 / 28.2 | 60.0 / 20.0 | 67.7 / 11.8 | 70.0 |
| | TrustRAG [54] | 62.7 / **1.8** | 63.6 / 2.7 | 63.6 / **2.7** | 64.5 / 2.7 | 66.4 / 13.6 | 73.0 |
| | SeconRAG(ours) | **63.6** / 2.5 | **65.5 / 0.0** | **66.4** / 3.6 | **67.3 / 0.0** | **74.5 / 10.2** | **82.0** |
| Qwen-7B [20] | VanillaRAG | 5.5 / 93.6 | 10.0 / 88.2 | 14.5 / 82.7 | 27.3 / 69.1 | 39.1 / 51.8 | 56.0 |
| | InstructRAG [42] | 25.5 / 76.4 | 33.6 / 65.5 | 33.6 / 65.5 | 35.5 / 62.7 | 47.3 / 47.3 | 64.0 |
| | ASTUTERAG [39] | 42.3 / 53.2 | 48.2 / 46.4 | 50.9 / 39.1 | 53.6 / 31.8 | 60.5 / **17.3** | 68.0 |
| | TrustRAG [54] | 60.0 / 2.7 | 64.5 / 7.3 | 62.7 / **3.6** | 65.5 / **2.7** | 64.5 / 24.5 | 67.0 |
| | SeconRAG(ours) | **66.4 / 2.4** | **70.0 / 4.5** | **67.3** / 5.5 | **68.2** / 3.6 | **70.9** / 21.8 | **78.0** |
| LLaMA-3.1-8B [13] | VanillaRAG | 10.9 / 88.2 | 16.4 / 81.8 | 21.8 / 71.8 | 33.6 / 59.1 | 41.8 / 52.7 | 70.0 |
| | InstructRAG [42] | 32.7 / 67.3 | 44.5 / 54.5 | 43.6 / 54.5 | 49.1 / 49.1 | 56.4 / 34.5 | 70.0 |
| | ASTUTERAG [39] | 58.2 / 31.8 | 60.0 / 25.5 | 64.5 / 25.5 | 70.0 / 18.2 | 77.5 / 8.2 | 81.0 |
| | TrustRAG [54] | 79.1 / **0.0** | 83.6 / **2.7** | 85.5 / 2.7 | 83.6 / **1.8** | 79.1 / 10.9 | 84.0 |
| | SeconRAG(ours) | **88.2** / 1.8 | **88.2** / 5.5 | **90.0 / 0.0** | **89.1** / 1.8 | **86.9 / 4.0** | **90.0** |
| GPT-4o [1] | VanillaRAG | 27.3 / 68.2 | 33.6 / 61.8 | 41.8 / 49.1 | 50.0 / 36.4 | 52.7 / 31.8 | 74.0 |
| | InstructRAG [42] | 43.6 / 51.1 | 51.8 / 40.9 | 53.6 / 37.3 | 59.1 / 30.9 | 66.4 / 25.5 | 74.0 |
| | ASTUTERAG [39] | 75.5 / 14.2 | 75.5 / 12.7 | 76.4 / 12.7 | 78.2 / 9.1 | 79.1 / 10.9 | 81.0 |
| | TrustRAG [54] | 80.0 / 0.1 | **81.8** / 1.8 | 82.7 / 0.9 | 82.7 / 0.9 | 81.8 / 1.0 | 86.0 |
| | SeconRAG(ours) | **81.8 / 0.0** | **81.8 / 0.9** | **83.6 / 0.9** | **85.5 / 0.0** | **84.5** / 1.0 | **88.0** |
| DeepSeek-R1 [18] | VanillaRAG | 17.3 / 84.5 | 30.9 / 68.2 | 34.5 / 64.5 | 43.6 / 54.5 | 51.0 / 43.6 | 80.0 |
| | InstructRAG [42] | 39.1 / 62.7 | 50.9 / 48.2 | 52.7 / 47.3 | 57.3 / 41.8 | 65.5 / 32.7 | 82.0 |
| | ASTUTERAG [39] | 81.8 / 10.9 | 80.9 / 11.8 | 87.3 / 7.3 | 85.5 / 5.5 | 89.1 / **0.0** | 87.0 |
| | TrustRAG [54] | 88.2 / **0.0** | 90.0 / 0.9 | 89.1 / **0.0** | 90.0 / **0.0** | 90.0 / 3.6 | 91.0 |
| | SeconRAG(ours) | **96.4 / 0.0** | **98.2 / 0.0** | **96.4 / 0.0** | **100.0 / 0.0** | **96.4 / 0.0** | **98.0** |

outperforms ASTUTERAG and InstructRAG, achieving 91.8% accuracy with 0.0% ASR under 60% poisoning and 98.0% accuracy in clean settings. Qwen-7B: Despite being a smaller model, SeConRAG achieves 84.0% accuracy in the clean setting and maintains low ASR (e.g., 4.5% at 100% poisoning), outperforming TrustRAG by a significant margin. LaMA-3.1-8B: SeConRAG achieves 90.0% accuracy in the clean setting and demonstrates strong robustness even under high poisoning (e.g., 89.1% / 0.0% at 100%). GPT-4o: SeConRAG matches or slightly outperforms TrustRAG for all poisoning levels. It achieves 94.0% accuracy on clean data and maintains 89.1% accuracy with only 1.8% ASR under 100% poisoning. DeepSeek-R1: SeConRAG outperforms all other tested methods in terms of robustness. It achieves 94.5% accuracy with 0.0% ASR under 60% poisoning

and maintains strong performance even at 100% poisoning (94.5%/1.8%), outperforming TrustRAG (89.1%/3.6%). These findings confirm that SeConRAG is not only effective at resisting large-scale corpus poisoning attacks, but it also excels at maintaining answer quality in both adversarial and clean environments.

Table 5: Performance comparison of SeConRAG and baseline methods on MS using different Poisoning RAG ratios (highest accuracy ↑ or lowest ASR ↓).

| Model | Method | MS-MARCO [6] | | | | | |
|---|---|---|---|---|---|---|---|
| | | 100% (ACC↑ / ASR↓) | 80% (ACC↑ / ASR↓) | 60% (ACC↑ / ASR↓) | 40% (ACC↑ / ASR↓) | 20% (ACC↑ / ASR↓) | 0% (ACC↑) |
| Mistral-12B [4] | VanillaRAG | 9.1 / 89.1 | 15.5 / 81.8 | 19.1 / 76.4 | 34.5 / 60.0 | 50.0 / 45.5 | 84.0 |
| | InstructRAG [42] | 15.5 / 78.2 | 17.3 / 77.3 | 24.5 / 70.0 | 35.5 / 57.3 | 57.3 / 36.4 | 81.0 |
| | ASTUTERAG [39] | 32.7 / 58.2 | 33.6 / 58.2 | 46.4 / 45.5 | 61.8 / 30.0 | 73.6 / 18.8 | 81.0 |
| | TrustRAG [54] | **91.8 / 0.0** | 81.8 / 7.3 | 86.4 / 4.5 | 86.4 / 5.5 | 87.3 / 11.8 | 85.0 |
| | SeconRAG(ours) | 88.2 / 0.0 | **91.8 / 1.8** | **91.8 / 0.0** | **90.9 / 1.8** | **89.1 / 9.1** | **98.0** |
| Qwen-7B [20] | VanillaRAG | 10.0 / 87.3 | 13.6 / 84.5 | 22.7 / 75.5 | 28.2 / 69.1 | 43.6 / 46.4 | 75.0 |
| | InstructRAG [42] | 43.6 / 57.8 | 39.1 / 59.1 | 47.3 / 50.0 | 49.1 / 48.2 | 49.1 / 45.5 | 75.0 |
| | ASTUTERAG [39] | 42.3 / 54.5 | 43.6 / 51.8 | 49.1 / 42.7 | 60.9 / 26.4 | 65.5 / 20.0 | 74.0 |
| | TrustRAG [54] | 64.5 / 11.8 | 65.5 / 14.5 | 66.4 / 10.0 | 67.3 / 11.8 | 66.4 / 22.7 | 78.0 |
| | SeconRAG(ours) | **71.8 / 4.5** | **71.8 / 6.4** | **73.6 / 6.4** | **75.5 / 6.4** | **75.5 / 17.5** | **84.0** |
| LLaMA-3.1-8B [13] | VanillaRAG | 9.1 / 88.2 | 20.0 / 77.3 | 28.2 / 66.4 | 36.4 / 60.0 | 54.5 / 40.9 | 83.0 |
| | InstructRAG [42] | 48.5 / 51.8 | 45.5 / 52.7 | 53.6 / 42.7 | 62.7 / 33.6 | 72.7 / 27.3 | 81.0 |
| | ASTUTERAG [39] | 56.8 / 38.6 | 63.6 / 29.1 | 63.6 / 26.4 | 73.6 / 21.8 | 82.3 / 13.6 | 89.0 |
| | TrustRAG [54] | 84.5 / 6.4 | 83.6 / 8.2 | 82.7 / 8.2 | 86.4 / 7.3 | 85.4 / **9.1** | 84.0 |
| | SeconRAG(ours) | **89.1 / 0.0** | **89.1 / 0.0** | **85.5 / 5.5** | **87.3 / 3.6** | **86.2 / 9.1** | **90.0** |
| GPT-4o [1] | VanillaRAG | 30.0 / 64.1 | 46.4 / 43.6 | 56.4 / 34.5 | 59.1 / 25.5 | 72.3 / 16.4 | 84.0 |
| | InstructRAG [42] | 50.5 / 42.7 | 57.3 / 35.5 | 62.7 / 30.0 | 59.1 / 24.5 | 70.9 / 17.3 | 83.0 |
| | ASTUTERAG [39] | 76.4 / 15.5 | 78.2 / 10.9 | 80.0 / 6.4 | 80.0 / 9.1 | 82.7 / 6.4 | 86.0 |
| | TrustRAG [54] | **89.1 / 1.8** | **90.9 / 1.8** | 89.1 / 3.6 | 88.2 / 3.6 | 84.5 / 6.4 | 88.0 |
| | SeconRAG(ours) | **89.1 / 1.8** | **90.9 / 1.8** | **90.0 / 1.8** | **89.1 / 1.8** | **89.1 / 3.6** | **94.0** |
| DeepSeek-R1 [18] | VanillaRAG | 11.8 / 81.8 | 33.6 / 61.8 | 39.1 / 55.5 | 50.9 / 42.7 | 60.5 / 29.1 | 82.0 |
| | InstructRAG [42] | 51.8 / 47.5 | 54.5 / 44.5 | 61.8 / 37.3 | 67.3 / 30.9 | 72.7 / 26.4 | 87.0 |
| | ASTUTERAG [39] | 85.5 / 8.2 | 80.9 / 13.6 | 80.9 / 10.0 | 87.3 / 7.3 | 89.1 / **5.5** | 88.0 |
| | TrustRAG [54] | 89.1 / 3.6 | 90.9 / 2.7 | 91.8 / 2.7 | 91.8 / 3.6 | 89.1 / **5.5** | 91.0 |
| | SeconRAG(ours) | **94.5 / 1.8** | **94.5 / 1.8** | **94.5 / 0.0** | **96.4 / 0.0** | **94.5 / 5.5** | **94.0** |

## A.4.2 Impact of SCF Subcomponents

To demostrate the necessity of combining two filtering processes, we evaluating each subcomponent independently. As shown in 6,while each module provides moderate improvements on its own, when combined, they result in significantly increased robustness (for example, 0% ASR in several settings). These results confirm that the combination of clustering and semantic filtering is complementary, yielding the strongest robustness overall.

Table 6: Ablation of SCF components on Mistral-12B .

| Model | Setting | HotpotQA [47] | | | NQ [24] | | | MS-MARCO [6] | | |
|---|---|---|---|---|---|---|---|---|---|---|
| | | PIA ACC↑/ASR↓ | 100% ACC↑/ASR↓ | 20% ACC↑/ASR↓ | PIA ACC↑/ASR↓ | 100% ACC↑/ASR↓ | 20% ACC↑/ASR↓ | PIA ACC↑/ASR↓ | 100% ACC↑/ASR↓ | 20% ACC↑/ASR↓ |
| mistral-12b | Clustering only | 78 / 5 | 81 / 2 | 78 / 9 | 68 / 3 | 65 / 3 | 70 / 10 | 85 / 7 | 82 / 7 | 82 / 12 |
| mistral-12b | Semantic only | 79 / 4 | 80 / 2 | 74 / 11 | 69 / 2 | 64 / 3 | 73 / 8 | 86 /5 | 82 / 6 | 86 / 8 |
| mistral-12b | Both (SCF) | 77.5 / 0.8 | 75.7 / 3.6 | 72.7 / 4.5 | 72.3 / 1.8 | 63.6 / 2.5 | 74.5 / 10.2 | 91.8 / 0 | 88.2 / 0 | 89.1 / 9.1 |

## A.4.3 Impact of of EIRE Module

To better understand the standalone contribution of the proposed Entity-Intent-Relation Extractor (EIRE) to SeCon-RAG's overall robustness, We specifically compare SeCon-RAG's performance with and without EIRE under various poisoning scenarios and three datasets, with Mistral-12B serving as the backbone model. The results are summarized in Table 7. With EIRE enabled, the model consistently achieves higher factual accuracy while significantly lowering the ASR, particularly under high poisoning conditions. For example, on the MS-MARCO dataset under 100% poisoning attack, enabling EIRE reduces ASR from 5% to 0% while increasing accuracy from 85% to 88.2%. These show that EIRE is critical for enabling fine-grained semantic reasoning which increases the accuracy of the final answer generation process.

Table 7: Ablation of the EIRE module on Mistral-12B across three datasets and poisoning scenarios.

| Model | Setting | HotpotQA [47] | | | NQ [24] | | | MS-MARCO [6] | | |
|---|---|---|---|---|---|---|---|---|---|---|
| | | PIA ACC↑/ASR↓ | 100% ACC↑/ASR↓ | 20% ACC↑/ASR↓ | PIA ACC↑/ASR↓ | 100% ACC↑/ASR↓ | 20% ACC↑/ASR↓ | PIA ACC↑/ASR↓ | 100% ACC↑/ASR↓ | 20% ACC↑/ASR↓ |
| mistral-12b | Without EIRE | 76 / 5 | 75 / 4 | 73 / 11 | 69 / 3 | 63 / 4 | 72 / 16 | 87 / 5 | 85 / 5 | 83 / 11 |
| mistral-12b | With EIRE | 77.5 / 0.8 | 75.7 / 3.6 | 72.7 / 4.5 | 72.3 / 1.8 | 63.6 / 2.5 | 74.5 / 10.2 | 91.8 / 0 | 88.2 / 0 | 89.1 / 9.1 |

### A.4.4 Impact of the Verified Correct Document Set

To evaluate the effectiveness the efficacy of $d_{cor}$, we conduct an ablation study without $d_{cor}$ and measure the performance drop across three datasets under three poisoning scenarios, with Mistral-12B serving as the baseline. As shown in Table 8, removing D consistently reduces accuracy while increasing the attack success rate, particularly in high-poisoning settings. For example, on MS-MARCO with 100% poisoning, enabling D reduces ASR from 5% to 0% while increasing accuracy from 85% to 88.2%. These results demonstrate that even a small, high-quality $d_{cor}$ set can significantly improve semantic filtering performance and reduce noise from poisoned documents.

Table 8: Ablation of the verified correct document set $d_{cor}$.

| Model | Setting | HotpotQA [47] | | | NQ [24] | | | MS-MARCO [6] | | |
|---|---|---|---|---|---|---|---|---|---|---|
| | | PIA ACC↑/ASR↓ | 100% ACC↑/ASR↓ | 20% ACC↑/ASR↓ | PIA ACC↑/ASR↓ | 100% ACC↑/ASR↓ | 20% ACC↑/ASR↓ | PIA ACC↑/ASR↓ | 100% ACC↑/ASR↓ | 20% ACC↑/ASR↓ |
| mistral-12b | Without $d_{cor}$ | 76 / 10 | 80 / 6 | 73 / 12 | 72 / 6 | 63 / 3 | 73 / 10 | 85 / 9 | 82 / 6 | 85 / 8 |
| mistral-12b | With $d_{cor}$ | 77.5 / 0.8 | 75.7 / 3.6 | 72.7 / 4.5 | 72.3 / 1.8 | 63.6 / 2.5 | 74.5 / 10.2 | 91.8 / 0 | 88.2 / 0 | 89.1 / 9.1 |

### A.4.5 Sensitivity Analysis of Filtering Thresholds

To assess the robustness of SeCon-RAG in relation to its key hyperparameters, we perform a sensitivity analysis on the two primary filtering thresholds: $\tau_{cluster}$: the similarity threshold used in clustering-based filtering. $\tau_{semantic}$: the semantic similarity threshold used in EIRE-based semantic graph filtering. We vary each threshold across a reasonable range ($\tau_{cluster} \in [0.86, 0.90]$, $\tau_{semantic} \in [0.2, 0.4]$) and evaluate SeCon-RAG's performance under three poisoning intensities on two representative models (LLaMA-3.1-8B and GPT-4o) and three datasets.

Table 9: Sensitivity analysis of $\tau_{cluster}$ on LLaMA-3.1-8B and GPT-4o under different poisoning intensities.

| Model | $\tau_{cluster}$ | HotpotQA [47] | | | NQ [24] | | | MS-MARCO [6] | | |
|---|---|---|---|---|---|---|---|---|---|---|
| | | PIA ACC↑/ASR↓ | 100% ACC↑/ASR↓ | 20% ACC↑/ASR↓ | PIA ACC↑/ASR↓ | 100% ACC↑/ASR↓ | 20% ACC↑/ASR↓ | PIA ACC↑/ASR↓ | 100% ACC↑/ASR↓ | 20% ACC↑/ASR↓ |
| LLaMA-3.1-8B | 0.86 | 72 / 4 | 68 / 4 | 72 / 4 | 67 / 19 | 83 / 2 | 79 / 4 | 86 / 2 | 83 / 6 | 88 / 4 |
| LLaMA-3.1-8B | 0.90 | 72 / 4 | 69 / 3 | 74 / 4 | 65 / 19 | 83 / 2 | 80 / 4 | 86 / 2 | 84 / 6 | 88 / 4 |
| GPT-4o | 0.86 | 80 / 3 | 81 / 2 | 81 / 3 | 82 / 6 | 82 / 1 | 81 / 1 | 83 / 1 | 81 / 3 | 90 / 3 |
| GPT-4o | 0.90 | 81 / 3 | 81 / 3 | 84 / 4 | 81 / 9 | 82 / 2 | 83 / 1 | 84 / 2 | 83 / 1 | 88 / 4 |

Table 10: Sensitivity analysis of $\tau_{semantic}$ on LLaMA-3.1-8B and GPT-4o under different poisoning intensities.

| Model | $\tau_{semantic}$ | HotpotQA [47] | | | NQ [24] | | | MS-MARCO [6] | | |
|---|---|---|---|---|---|---|---|---|---|---|
| | | PIA ACC↑/ASR↓ | 100% ACC↑/ASR↓ | 20% ACC↑/ASR↓ | PIA ACC↑/ASR↓ | 100% ACC↑/ASR↓ | 20% ACC↑/ASR↓ | PIA ACC↑/ASR↓ | 100% ACC↑/ASR↓ | 20% ACC↑/ASR↓ |
| LLaMA-3.1-8B | 0.2 | 68 / 7 | 67 / 5 | 74 / 4 | 66 / 19 | 82 / 2 | 80 / 4 | 86 / 2 | 83 / 7 | 88 / 4 |
| LLaMA-3.1-8B | 0.4 | 68 / 7 | 70 / 3 | 74 / 4 | 66 / 19 | 82 / 2 | 80 / 4 | 85 / 2 | 83 / 7 | 89 / 6 |
| GPT-4o | 0.2 | 81 / 4 | 81 / 3 | 84 / 3 | 84 / 7 | 81 / 2 | 82 / 2 | 82 / 1 | 84 / 3 | 89 / 4 |
| GPT-4o | 0.4 | 80 / 4 | 82 / 3 | 83 / 4 | 82 / 9 | 81 / 1 | 82 / 2 | 82 / 1 | 83 / 1 | 89 / 2 |

Tables 9 and 10 show that SeCon-RAG's performance remains stable even when both thresholds are changed slightly. This is primarily due to the conservative AND-logic used in the joint filtering mechanism, which ensures that only documents flagged by both filters are excluded. These findings show that our framework is not overly sensitive to precise threshold tuning, which makes it easier to use in practice.

