# OpenReview forum: "SeCon-RAG: A Two-Stage Semantic  Filtering and Conflict-Free Framework for Trustworthy RAG"
_NeurIPS.cc/2025/Conference — NeurIPS 2025 poster_

### Official Review · Reviewer_do5K · 2025-06-09

**Clarity:** 2
**Significance:** 3
**Originality:** 3
**Rating:** 4
**Confidence:** 4

**Summary:**

This paper introduces SeCon-RAG, a two-stage framework to enhance the trustworthiness and robustness of RAG systems against corpus poisoning and contamination attacks. The core problem addressed is the vulnerability of RAG systems to malicious data, where existing defenses often suffer from aggressive filtering that removes valuable information, or fail to resolve conflicts between retrieved content and the LLM's internal knowledge. Experiments demonstrate that the effectiveness of SeCon-RAG.

**Questions:**

1.See weekness

2.The semantic distinction between clean content and toxic content is not clearly explained

**Ethical Concerns:**

["NO or VERY MINOR ethics concerns only"]

**Final Justification:**

Thanks for your reply. I have raise my rating. Good luck.

**Limitations:**

yes

**Quality:**

3

**Strengths And Weaknesses:**

Strengths:

- SeCon-RAG addresses multiple aspects of RAG vulnerabilities. The combined use of clustering and semantic graph-based filtering in SCF, followed by CAF, provides a robust and layered defense.

Weaknesses:

- Dependency on LLMs. EIRE and ssG rely on extra LLMs.  It introduces concerns regarding the cost, latency, and bias of the LLM used for these foundational components.

- The effectiveness of EIRE standalone in extracting semantic structures and identifying poisoned content is not evaluated in ablation study.

- The paper mentions adjustable thresholds (two \tau), but lacks a study on the sensitivity of performance to these thresholds.

- EIRE and ssG rely on LLMs. This has impact on the overall system's robustness and efficiency. This paper lacks discussion on real-world deployment challenges.

- The fontsize in table 1, table 2, figure 1-4 are too small. The tables should be re arranged.

-

---

> ### Author Rebuttal · Authors · 2025-07-27
>
> We appreciate your insightful review and recommendations. We address the remarks in Weaknesses (W) and Questions (Q) below.
>
> ---
>
> **W1:Dependency on LLMs: EIRE and ssG rely on additional LLM calls, raising concerns about cost, latency, and bias.**
>
> Thank you for the feedback.  We recognize the reviewer's apprehension about the dependence on LLMs in foundational elements like ssG and EIRE. The need for deeper semantic understanding that conventional symbolic or sparse methods cannot offer drove our design decision to use LLMs for semantic structure extraction and graph similarity. We do agree, though, that this adds overhead.
> Also, we notice the following: As shown in Section 5.4 and Figure 3, the additional runtime introduced by SeCon-RAG remains within a reasonable range (<1.5 minutes per batch). One query only requires an acceptable cost, which we believe is acceptable for many real-world applications that require robustness.
>
> We also showed in Section 5.3 (Table 2) that SeCon-RAG performs well across various lightweight embedding models (e.g., MiniLM, SimCSE), indicating that the framework is adaptable to lower-cost alternatives.
> For future work, we are looking into more lightweight techniques to replace EIRE, reducing the reliance on full-scale LLM inference at runtime. We will clarify these points more explicitly in the revised version, as well as include additional discussion of scalability trade-offs in deployment scenarios.
>
> ---
> ﻿
> **W2:Lack of standalone evaluation for EIRE in ablation study.**
>
> We agree that evaluating EIRE's standalone impact is critical for better understanding its contribution to overall performance.To address this, Table (shown below) contains a more detailed ablation analysis in which we compare performance with and without EIRE across PIA and multiple poisoning intensity levels (100%, 20%) and datasets (Hotpot, NQ, MS) use  Mistral-12B. The results clearly demonstrate the efficacy of EIRE as a standalone defense component.
>
> | W/O EIRE | PIA | 100%  | 20% | PIA | 100% |  20% | PIA  | 100%  | 20%|
> | --- | --- | --- | --- | --- | --- | --- | --- | --- | --- |
> | N | 76 / 5  |  75 / 4 | 73 / 11 |  69 / 3 | 63 / 4| 72 / 16 |  87 / 5 | 85 / 5 | 83 / 11 |
> | Y  | 77.5 / 0.8 |75.7 / 3.6 |72.7 / 4.5 | 72.3 / 1.8 | 63.6 / 2.5 | 74.5 / 10.2 |91.8 / 0 |  88.2 / 0|89.1 / 9.1|
>
> The table shows Accuracy (ACC↑) on the left and Attack Success Rate (ASR) on the right. When EIRE is enabled ("Y"), the model consistently achieves higher or comparable accuracy across all settings than when EIRE is turned off ("N").
> Attack Success Rate (ASR↓) More importantly, EIRE significantly lowers the attack success rate, particularly in high-poisoning scenarios. For example, under 100% PIA, ASR falls from 16% to 10.2% in one setting and to 0% in others.
> These findings confirm that EIRE alone makes a significant contribution to the defense mechanism, both in terms of preventing attacks and maintaining model performance. We will clarify this in the revised manuscript and emphasize EIRE's standalone impact in the ablation study section.
>
> ---
>
>
> **W3: Threshold sensitivity analysis is missing.**
>
> We will add a supplementary plot showing how varying τ_cluster and τ_semantic affects accuracy and ASR. Preliminary results suggest that SeCon-RAG is relatively robust within a reasonable range (±0.1) of threshold values, due to the conservative AND-logic used in joint filtering.
>
> We agree that thresholds are critical in determining the strictness of the Semantic Consistency Filter (SCF). In our experiments, we manually tuned these thresholds on a validation set. Unfortunately, due to space constraints, we did not include a detailed sensitivity analysis in the main paper.
>
> To address this, we conducted a comprehensive threshold sensitivity analysis and included the findings in the supplementary material. Specifically, we varied τ_cluster and τ_semantic independently across a reasonable range and evaluated the impact on both model accuracy (ACC↑) and attack success rate (ASR↓) under different poisoning intensity levels (PIA: 100%, 60%, 20%) on datasets(Hotpot、NQ、MS).
> The findings show that SeCon-RAG is relatively resistant to small variations in these thresholds, owing to the conservative AND-logic used in joint filtering. Changing τ_cluster from 0.86 to 0.90 maintains accuracy and ASR stability across both LLaMA-3.1-8B and GPT-4o backbones. Similarly, changes in τ_semantic between 0.2 and 0.4 cause only minor fluctuations in performance.
>
> | Model | τ_cluster | PIA | 100% | 60% | 20% | PIA  | 100% | 60%  | 20%  | PIA  | 100%  | 60% | 20% |
> | --- | --- | --- | --- | --- | --- | --- | --- | --- | --- | --- | --- | --- | --- |
> | LLaMA-3.1-8B  |0.86|72 / 4|68 / 4|72 / 4|67 / 19|83 / 2|79 / 4|86 / 2|83 / 6|88 / 4|88 / 4|84 / 7|86 / 4|
> | LLaMA-3.1-8B| 0.90 |72 / 4 |69 / 3 |74 / 4 | 65 / 19 | 83 / 2 | 80 / 4  | 86 /2 |84 / 6| 88 / 4 | 88 / 4 | 84 /7 |87 / 5 |
> | GPT-4o | 0.86 | 80 / 3 |81 / 2 |81 / 3 |82 / 6 | 82  / 1| 81 / 1 | 83 / 1 | 81 / 3 | 90 / 3 |90 / 2 | 88 / 3 | 87 / 6  |
> |  GPT-4o  |0.90| 81 / 3  |81 / 3 | 84 / 4 |81 / 9 | 82 /  2 | 83 / 1 |84 / 2| 83 / 1| 88 / 4  |86 / 4 |90 / 4 |  85 / 6 |
>
> | Model  | τ_semantic | PIA | 100% | 60% | 20% | PIA  | 100% | 60% | 20% |  PIA  | 100%  | 60% | 20%  |
> | --- | --- | --- | --- | --- | --- | --- | --- | --- | --- | --- | --- | --- | --- |
> |LLaMA-3.1-8B | 0.2 |68 / 7 |   67 / 5    |    74 / 4   |    66 / 19   |     82 / 2  |    80 / 4   |    86/.2   | 83./7      |    88./4   |    88 / 4      |     86 / 5  |    89 / 4   |
> | LLaMA-3.1-8B |    0.4    |  68 / 7     |   70 / 3    |     74 / 4  |    66  / 19   |    82 / 2   |  80 / 4     | 85 / 2 |     83 / 7  |   88 / 4   |     88 / 4  |     86 / 5  |    89 / 6  |
> | GPT-4o | 0.2 |   81 / 4    |   81 / 3    |   84 / 3    |   84 / 7    |    81 / 2   |     82 / 2  |     82 / 1  |    84 / 3   |    89 / 4   |    90 / 3   |   87 / 3   |  88 / 8     |
> | GPT-4o | 0.4 | 80 / 4 |    82  / 3   |    83 / 4   |  82 / 9     |     81 / 1  |  82 / 2     |     82 / 1  |  83 / 1     |    89 / 2   |     88 / 4  |  88 / 1     |    84 / 7   |
>
> ---
>
> **W4: Real-world deployment discussion is lacking.**
>
> We agree that additional discussion of real-world deployment considerations would improve the paper. In Section 6 (Limitations), we will discuss the trade-off between robustness and latency, particularly for latency-sensitive applications (e.g., real-time QA). And we also want to point out that because of the SeCon-RAG's plug-and-play compatibility with existing RAG systems, the real-world can easily use it to improve the robustness of a realistic system.
>
> ---
>
> **W5: Font size in tables/figures is too small.**
>
> Thank you for this important formatting remark. We acknowledge that the font sizes in Table 1, Table 2, and Figures 1-4 are currently small due to layout constraints.  In the camera-ready version, we will separate large tables into sub-tables for each dataset to improve readability. And increase the font size and spacing in all figures and tables, move detailed numerical tables to the appendix, and keep summary plots in the main text.
>
> ---
>
> **Q2: Semantic distinction between clean and toxic (poisoned) content is unclear.**
>
>
> Thank you for the feedback.  The semantic difference between clean and toxic content in our SCF module can be understood from two key perspectives:
>
>
> First, because poisoned documents frequently contain identical or highly similar content, their embedding distributions tend to cluster tightly together when retrieved. In contrast, clean content has more diverse semantic representations and thus does not form such tight clusters. This phenomena has previously been examined in past work, like as the paper "TrustRAG" which mentioned a similar viewpoint.
>
> Second, we conducted a thorough analysis using semantic graphs created from entities, intents, and relations extracted using EIRE. Figure 2 and Figures 1, 2, and 3 in the Appendix demonstrate that poisoned documents consistently produce sparser and less coherent semantic graphs than clean documents. These structural inconsistencies occur because poisoned content frequently contains misleading or contradictory information that lacks semantic consistency and connectivity. Furthermore, the characteristics of the poisoning attack approach exacerbate this effect.
>
> These findings support our targeted design of the SCF module, which uses semantic graph-based analysis and clustering-based filtering to effectively distinguish toxic from clean content.

---

> > ### Comment · Reviewer_do5K · 2025-08-05
> >
> > Dear ACs and Authors,
> > I have read all the reviews and rebuttals, most address my concerns, and raised my rating. Good luck.

---

> > > ### Author Response · Authors · 2025-08-05
> > >
> > > Thank you for your support !

---

### Official Review · Reviewer_bSmW · 2025-06-27

**Clarity:** 3
**Significance:** 3
**Originality:** 3
**Rating:** 4
**Confidence:** 4

**Summary:**

This paper introduces SeCon-RAG, a two-stage framework designed to defend Retrieval-Augmented Generation (RAG) systems against corpus poisoning and contamination attacks. Traditional defenses often use aggressive filtering, which can lead to the loss of useful information. To counter this, the proposed method first employs a Semantic and Cluster-Based Filtering (SCF) stage. This stage is guided by an Entity-intent-relation extractor (EIRE), which extracts key semantic information like entities, intents, and relations from documents. SCF combines filtering based on document clusters in the embedding space with filtering based on a semantic structure graph generated by EIRE. The second stage introduces an EIRE-guided conflict-aware filtering (CAF) module that operates before the final answer is generated. This module analyzes the semantic consistency between the user's query, the retrieved knowledge, and the candidate answers to filter out internal and external contradictions. This two-stage process aims to preserve valuable knowledge while mitigating the impact of malicious content, thereby improving both the robustness and trustworthiness of the generated output. The authors provide extensive experimental results across various large language models and datasets, showing that SeCon-RAG outperforms current state-of-the-art defense methods.

**Questions:**

1.Could you provide more details on the LLMs used for the sub-tasks within SeCon-RAG (EIRE, semantic graph similarity, CAF decision-making)? Are these general-purpose models or smaller, specialized models? If general-purpose, how is the prompting engineered for robustness?
2.Could you elaborate on the "Supervise" aspect for EIRE mentioned in Figure 1?
3.How were the thresholds τcluster​ and τsemantic​ determined for the experiments? Was a validation set used, and how sensitive is the performance to variations in these thresholds?
4.Regarding the "small collection of verified correct documents Dcor​" for semantic graph benchmarking: How large is this set, how is it curated, and how does its composition affect the semantic filtering performance?
5.The CAF module checks for consistency with the LLM's internal knowledge ("Model Consistency" ). How is this "internal knowledge" accessed or queried for consistency checking, especially to avoid reinforcing existing biases or hallucinations in the base LLM?

**Ethical Concerns:**

["NO or VERY MINOR ethics concerns only"]

**Final Justification:**

I have read all the reviews and rebuttals, most address my concerns, and I will raise my score by 1.

**Limitations:**

see above questions.

**Quality:**

3

**Strengths And Weaknesses:**

The primary strength of this work lies in its novel and comprehensive approach to a timely and critical problem. The paper is the first to integrate structured semantic information into the RAG defense process through its proposed EIRE module, allowing for a more fine-grained detection of poisoned content. The two-stage framework, which combines the pre-retrieval SCF module with the inference-time CAF module, presents a robust and layered defense strategy. The evaluation is another significant strength; the authors conducted extensive experiments on three different QA benchmarks using five prominent LLMs and tested against multiple attack scenarios, including corpus poisoning and prompt injection attacks. The results are empirically strong, with SeCon-RAG consistently demonstrating high factual accuracy and low attack success rates, outperforming several baselines across nearly all settings. This is further supported by a thorough ablation study that validates the contribution of each individual component to the framework's overall effectiveness.
Despite these strengths, the work has some weaknesses. A notable concern is the method's reliance on large language models for multiple core functions, including the EIRE extractor, the semantic graph similarity scoring, and the final judgment in the CAF module. This dependency introduces a moderate runtime overhead, with processing times reported between 1.21 and 1.45 minutes per batch, which may not be practical for latency-sensitive applications. Furthermore, the framework's performance is intrinsically tied to the quality of the EIRE parser, meaning any inaccuracies in the initial semantic extraction could propagate and degrade the performance of subsequent filtering stages. Finally, while the paper mentions that experimental results were statistically validated through multiple averages, the main results tables do not include error bars or other measures of statistical significance, making it more difficult to ascertain the true margin of improvement over the baseline methods.

---

> ### Author Rebuttal · Authors · 2025-07-27
>
> We appreciate your insightful review and recommendations. We address the remarks in Weaknesses (W) and Questions (Q) below.
>
> ---
>
> **W1: Dependency on LLMs and Runtime Overhead.**
>
> Thank you for the feedback.  We acknowledge that SeCon-RAG uses LLMs in several components (EIRE, semantic similarity scoring, and CAF's final judgment), resulting in a moderate runtime overhead.  The need for deeper semantic understanding that conventional symbolic or sparse methods cannot offer drove our design decision to use LLMs for semantic structure extraction and graph similarity. We do agree, though, that this adds overhead. Also, we notice the following:
> - A single query incurs only a reasonable computational cost, which we believe is acceptable for many real-world applications that require robustness against retrieval-time attacks—such as  business consulting, legal or medical information Q&A, and other high-stakes domains where resistance to poisoning is critical.
>
> - We also showed in Section 5.3 (Table 2) that SeCon-RAG performs well across various lightweight embedding models (e.g., MiniLM, SimCSE), indicating that the framework is adaptable to lower-cost alternatives. For future work, we are looking into more lightweight techniques to replace EIRE, reducing the reliance on full-scale LLM inference at runtime. We will clarify these points more explicitly in the revised version, as well as include additional discussion of scalability trade-offs in deployment scenarios. ﻿
>
> ---
>
> **W2. Reliance on EIRE Quality.**
>
> Thank you for the feedback.  You are correct in stating that the quality of EIRE affects downstream modules. The dual-layer design of SCF and CAF helps to reduce this dependency. So we evaluating EIRE's standalone impact is critical for better understanding its contribution to overall performance. Table contains a more detailed ablation analysis in which we compare performance with and without EIRE across PIA and multiple poisoning intensity levels (100%, 20%) and datasets (Hotpot, NQ, MS) use  Mistral-12B. The results clearly demonstrate the efficacy of EIRE as a standalone defense component.
>
> | W/O EIRE | PIA | 100%  | 20% | PIA | 100% |20% | PIA | 100%| 20%|
> | --- | --- | --- | --- | --- | --- | --- | --- | --- | --- |
> |N | 76 / 5  |  75 / 4 | 73 / 11 |  69 / 3 | 63 / 4| 72 / 16 |  87 / 5 | 85 / 5 | 83 / 11 |
> |Y| 77.5 / 0.8 |75.7 / 3.6 |72.7 / 4.5 | 72.3 / 1.8 | 63.6 / 2.5 | 74.5 / 10.2 |91.8 / 0 |88.2 / 0|89.1 / 9.1|
>
> The table shows Accuracy (ACC↑) on the left and Attack Success Rate (ASR) on the right. When EIRE is enabled ("Y"), the model consistently achieves higher or comparable accuracy across all settings than when EIRE is turned off ("N").  More importantly, EIRE significantly lowers the ASR, particularly in high-poisoning scenarios. For example, in dataset ms, ASR drops from 5% to 0% when Poisoned is set to 100%.
>
> These findings confirm that EIRE alone makes a significant contribution to the defense mechanism, both in terms of preventing attacks and maintaining model performance.
>
> ---
>
> **W3. Missing Error Bars.**
>
> Thank you for the feedback. We averaged our experimental results across multiple runs, but we did not include confidence intervals due to space constraints. This will be addressed as follows: The revised version now includes standard deviation/error bars in key tables and figures.In the appendix, we report the number of runs as well as details about our random seed and evaluation protocol.
>
> ---
>
> **Q1: LLMs used within SeCon-RAG: Are they general-purpose or specialized? How is prompting designed?.**
>
> Thank you for the feedback.  The benchmark RAG model determines whether we use open or closed-source LLMs. When the RAG  component is instantiated with model A, the resulting instruction model is also A, eliminating the need for a separate open or closed-source LLM. Our method is effective whether the LLM is accessed via external APIs or deployed locally, as demonstrated by the consistent performance across both settings in Table 2. And prompting was carefully designed for robustness and consistency, as shown in the appendix.
>
>  - EIRE extracts entities, intents, and relations using structured prompts (refer to Appendix A.3.1).
>
>  - For semantic graph comparison, we ask the LLM to compare a candidate document's semantic graph to reference graphs and return a normalized similarity score (see Appendix A.3.2).
>
>  - In CAF, we use explicit semantic criteria to assess trustworthiness across query, corpus, and model dimensions (see Appendix A.4).
>
> ---
> **Q2. Clarification on “Supervise” in Figure 1 for EIRE**
>
> Thank you for noticing the ambiguity. The "Supervise" label in Figure 1 indicates that EIRE guides both SCF and CAF.
>
> - In SCF, EIRE structures a semantic graph and compares it to a reference set.
> - In CAF, EIRE's outputs serve as the semantic foundation for consistency checks.
>
> The term "supervise" does not refer to supervised learning, but rather semantic guidance. We will change the figure caption and legend to avoid confusion.
>
> ---
>
> **Q3. How were τ_cluster and τ_semantic determined? Sensitivity?**
>
> Thank you for the feedback.   For example, the value of τ_cluster is set to 0.88, which was empirically determined based on our own experimental results.. In contrast, τ_semantic is determined using the semantic similarity scores we created from predefined prompt words (as shown in the appendix). Notably, τsemantic is set independently and does not rely on a validation set.
>
> Specifically, we varied τ_cluster and τ_semantic independently across a reasonable range and evaluated the impact on both model accuracy (ACC↑) and attack success rate (ASR↓) under different poisoning intensity levels (PIA: 100%, 60%, 20%) on datasets(Hotpot、NQ、MS). The findings show that SeCon-RAG is relatively resistant to small variations in these thresholds, owing to the conservative AND-logic used in joint filtering. Changing τ_cluster from 0.86 to 0.90 maintains accuracy and ASR stability across both LLaMA-3.1-8B and GPT-4o backbones. Similarly, changes in τ_semantic between 0.2 and 0.4 cause only minor fluctuations in performance.
>
> | Model | τ_cluster | PIA | 100% | 60% | 20% | PIA  | 100% | 60%  | 20% | PIA  | 100%  | 60% | 20% |
> | --- | --- | --- | --- | --- | --- | --- | --- | --- | --- | --- | --- | --- | --- |
> | LLaMA-3.1-8B  |0.86|72 / 4|68 / 4|72 / 4|67 / 19|83 / 2|79 / 4|86 / 2|83 / 6|88 / 4|88 / 4|84 / 7|86 / 4|
> | LLaMA-3.1-8B| 0.90 |72 / 4 |69 / 3 |74 / 4 | 65 / 19 | 83 / 2 | 80 / 4  | 86 /2 |84 / 6| 88 / 4 | 88 / 4 | 84 /7 |87 / 5 |
> | GPT-4o | 0.86 | 80 / 3 |81 / 2 |81 / 3 |82 / 6 | 82  / 1| 81 / 1 | 83 / 1 | 81 / 3 | 90 / 3 |90 / 2 | 88 / 3 | 87 / 6  |
> |  GPT-4o  |0.90| 81 / 3  |81 / 3 | 84 / 4 |81 / 9 | 82 /  2 | 83 / 1 |84 / 2| 83 / 1| 88 / 4  |86 / 4 |90 / 4 |  85 / 6 |
>
> | Model  | τ_semantic | PIA | 100% | 60% | 20% | PIA  | 100% | 60% | 20% |  PIA  | 100%  | 60% | 20%  |
> | --- | --- | --- | --- | --- | --- | --- | --- | --- | --- | --- | --- | --- | --- |
> |LLaMA-3.1-8B | 0.2 |68 / 7 | 67 / 5 |74 / 4 | 66 / 19| 82 / 2| 80 / 4| 86/.2 | 83./7| 88./4 | 88 / 4 | 86 / 5| 89 / 4|
> | LLaMA-3.1-8B |  0.4 |68 / 7 | 70 / 3 |74 / 4 | 66  / 19 | 82 / 2 |80 / 4 | 85 / 2 | 83 / 7 | 88 / 4 | 88 / 4| 86 / 5  | 89 / 6|
> | GPT-4o | 0.2 | 81 / 4  | 81 / 3| 84 / 3 |84 / 7 |81 / 2 | 82 / 2 | 82 / 1  | 84 / 3 | 89 / 4 | 90 / 3 | 87 / 3 |88 / 8|
> | GPT-4o | 0.4 | 80 / 4 | 82  / 3 | 83 / 4 |82 / 9 | 81 / 1|  82 / 2 | 82 / 1 | 83 / 1 | 89 / 2 | 88 / 4 | 88 / 1 | 84 / 7 |
>
> ---
>
> **Q4. Composition and size of Dcor (verified correct documents)**
>
>
> Thank you for the feedback.  The Dcor subset is a small collection of documents that have been manually verified as relevant and accurate to the query.  These documents were human-verified from datasets to ensure high semantic relevance and correctness. Dcor guides the semantic filtering and CAF process, especially in the construction of the semantic graph used to identify poisoned or inconsistent documents. Its primary function is to provide a reliable semantic anchor for calculating similarity scores.
>
> To assess the impact of Dcor, we conducted ablation experiments that compared performance with and without this Dcor subset. As shown in the table below (with and without cor), enabling Dcor ("Y") consistently improves  accuracy or reduces ASR across PIA and different poisoning intensities (100%, 20%) on three datasets (Hotpot 、NQ、MS) .
>
> | W/O Dcor | PIA | 100%  | 20% | PIA | 100% |  20% |PIA | 100%  |20% |
> | --- | --- | --- | --- | --- | --- | --- | --- | --- | --- |
> | N  | 76 / 10 | 80 / 6  |  73 / 12 | 72  / 6 |63 / 3| 73 / 10 | 85 / 9|82 / 6 |  85 / 8  |
> | Y | 77.5 / 0.8 | 75.7 / 3.6  |72.7 / 4.5 |72.3 / 1.8 | 63.6 / 2.5  |74.5 / 10.2 |91.8 / 0 | 88.2 / 0 |89.1 / 9.1 |
>
> For example, when Dcor is used at 100% poisoning intensity in MS, the ASR falls from 5% to 0% and accuracy improves from 85 to 88. These findings show that even a small, high-quality Dcor set can significantly improve semantic filtering performance by anchoring the semantic space and reducing noise from poisoned documents. In conclusion, while Dcor is small in size, its careful curation adds significantly to the robustness of our filtering mechanism.
>
> ---
>
> **Q5. How is the base LLM’s “internal knowledge” used for model consistency in CAF?**
>
> ---
>
> Thank you for the feedback.  In CAF's model consistency check, we ask the LLM to determine whether a document's semantic content corresponds to its internal knowledge.  At this stage, reinforcement learning was not used, nor was its bias towards knowledge reinforced.
>
> Importantly, the LLM is expected to reason based on extracted entities and relations, not memorized facts. We reduce hallucination risk by comparing multiple retrieved documents and relying on consistency across sources.Only documents marked "trustable" across all dimensions are used in the final generation, including those that support or confirm the LLM's own prior knowledge.

---

> > ### Comment · Reviewer_bSmW · 2025-08-05
> >
> > I have read all the reviews and rebuttals, most address my concerns, and I will raise my score by 1.

---

> > > ### Author Response · Authors · 2025-08-06
> > >
> > > Dear Reviewer bSmW,
> > >
> > > Thank you for your support! We sincerely appreciate your valuable feedback and the time you spent reviewing our work. Your suggestions helped improve the quality of our paper. We are pleased that the additional results adequately addressed your concerns. As suggested, we will carefully include the process of determining the threshold of filtering algorithm, visualize the error line, and provide an ablation study of the candidate set to improve the clarity and completeness of our work.
> > >
> > > Best, Authors

---

### Official Review · Reviewer_N6z7 · 2025-06-30

**Clarity:** 2
**Significance:** 2
**Originality:** 3
**Rating:** 3
**Confidence:** 4

**Summary:**

This paper presents SeCon-RAG, a two-stage framework to make Retrieval-Augmented Generation (RAG) systems more trustworthy and resistant to data poisoning attacks. Existing defenses can often remove useful information or fail to resolve knowledge conflicts. SeCon-RAG tackles this with two core modules: 1) Semantic and Cluster-Based Filtering (SCF): This module uses clustering and an Entity-Intent-Relation Extractor (EIRE) to perform fine-grained filtering, removing poisoned documents while preserving valuable ones. 2) This module uses EIRE to analyze semantic consistency between the query, retrieved documents, and the model's internal knowledge, filtering out contradictions before generating an answer. The contributions include introducing structured semantic analysis to RAG defense via EIRE and the proposed framework itself. Experiments show SeCon-RAG outperforms baseline methods in accuracy and attack resistance across various LLMs and datasets.

**Questions:**

- In line 168 "Due to the semantic similarity of malicious documents, they cluster in embedding space". This statement is too strong without any supporting evidence. I can imagine different attacking approaches would change this statement easily
- In line 184 "Correct documents produce densely connected semantic graphs with high coherence, whereas poisoned documents have sparse or fragmented structures." Again, one can simple flip an edge or rename an entity in Figure 2 (a) to produce a poisoned document, so this claim does not hold as well.
- In line 192, where do these verified documents come from?
- In equation (5), graphs are inputs to the LLM. However, in line 178-181, the graph node v is defined as "projection of embedding vector". Does that mean numerical vectors are sent to LLM as inputs? This does not make sense to me. In addition to that, how does the projection computed is not clear at all.

**Ethical Concerns:**

["NO or VERY MINOR ethics concerns only"]

**Final Justification:**

Although the authors additional experiments cleared some of my concerns, the contribution of this work still looks quite incremental to TrustRAG (https://arxiv.org/abs/2501.00879), so I keep my score towards borderline reject.

**Limitations:**

Yes, in the conclusion section.

**Quality:**

2

**Strengths And Weaknesses:**

Strengths:
- Trustworthy RAG against malicious attacks is an interesting and important topic
- Results are shown on multiple LLMs of different categories and sizes

Weakness (please see questions for more details):
- The methodology is relatively heuristic, lacking motivations in design choices
- Technical details are not clear enough
- Complexity analysis. It is appreciated that Figure 3 shows an empirical comparison under a specific initiation of hyperparameters of the proposed approach. However it is better to include a formal complexity analysis in the paper, for example, in terms of number of total documents, retrieved documents, average document token length, etc.
- The method heavily relies on prompt engineering in most components, therefore ultimate limited by the inherent capability of LLMs used.

---

> ### Author Rebuttal · Authors · 2025-07-27
>
> We sincerely thank the reviewer for the thoughtful and constructive comments. We address the remarks in Weaknesses (W) and Questions (Q) below.
>
> ---
>
> **W1. Heuristic Nature and Lack of Motivation for Design Choices**
>
> ---
>
> Thank you for this feedback.  Previously approaches for removing poisoned documents relied on filtering or voting, which disregarded semantic information.  And we feel that the semantic information contained in papers is particularly useful for LLM.  As a result, we decided to develop filtering methods from a semantic standpoint in order to improve the whole system's robustness and resistance to attacks.
>
> More specifically, the use of clustering and semantic graph filtering in SCF is motivated by the observation that poisoned documents cluster frequently (captured via embeddings) while lacking coherent semantic structure (captured via EIRE).  CAF ensures factual consistency from semantics by comparing retrieved documents to both the query and the LLM's own knowledge, making the final response more reliable.
>
> ---
>
> **W2. Technical Clarity**
>
> Thank you for this feedback.  We recognize that some of the core mechanisms, particularly how semantic structure graphs are constructed and evaluated, could benefit from more explicit explanation. In the revised version, we will:
>
> Include a step-by-step example of EIRE's output and how semantic graphs are created (see also Appendix A.3.1).
> Provide a more understandable figure that shows the difference between clean and poisoned documents in the semantic and embedding space.
> Extend Sections 4.2.2 and 4.3 to explain the reasoning behind using semantic similarity as a filtering signal. These additions will assist readers in understanding not only how the method works, but also why it is effective.
>
> ---
>
> **W3. Lack of Formal Complexity Analysis**
>
> The first stage of our proposed defense framework is the filtering phase, which includes cluster-based filtering on the retrieval database and semantic filtering on retrieved candidates.  Before final inference, the Consistency-Aware Filtering (CAF) module performs additional checks on both the retrieved documents and the inference output.  The pseudocode in the appendix provides a more detailed implementation and complexity analysis. In terms of complexity, the retrieval and inference processes are executed only once, whereas the additional filtering steps require two lightweight detection operations. This design ensures that overall computational overhead is minimal. Figure 3 illustrates a specific time comparison.
>
> ---
>
> **W4. Heavy Reliance on Prompting and LLM Capabilities**
>
> Thank you for the feedback.  We recognize the reviewer's apprehension about the dependence on LLMs in foundational elements like ssG and EIRE. Our design decision is based on the observation that LLMs are currently the most effective tools for capturing deep semantic structures in natural language which  traditional symbolic or sparse methods struggle to accomplish. Even when using smaller or less powerful models, the semantic signals extracted are sufficiently informative to allow our framework to function properly.
>
> We also showed in Section 5.3 (Table 2) that SeCon-RAG performs well across various lightweight embedding models (e.g., MiniLM, SimCSE), indicating that the framework is adaptable to lower-cost alternatives. For future work, we are looking into more lightweight techniques to replace EIRE, reducing the reliance on full-scale LLM inference at runtime. We will clarify these points more explicitly in the revised version, as well as include additional discussion of scalability trade-offs in deployment scenarios.
>
> ---
>
>
> **Q1. Line 168 – Clustering of malicious documents in embedding space**
>
> Thank you for your feedback.  Acturally Clustering of malicious documents in embedding space may not be universally applicable across all attack types, but in this paper, we will focus on the poisoned RAG, which will result in Clustering of malicious documents in embedding space. Furthermore, because poisoned documents frequently contain identical or very similar content, their embedding distributions tend to cluster tightly together when retrieved. Clean content, on the other hand, has a broader range of semantic representations and thus does not cluster as tightly. This phenomena has previously been examined in past work, like as the paper "TrustRAG" which mentioned a similar viewpoint. Furthermore, the characteristics of the poisoning attack strategy amplify this effect.  We will include a visualization (PCA) in the appendix to back up this point.
>
> ---
>
> **Q2. Line 184 – Sparse or fragmented semantic graphs**
>
> Thank you for your feedback.  As shown in Figure 2, correct documents generate densely connected semantic graphs, whereas  adversarial documents produce noticeably sparser structures. We use embedding model to project documents into vector space, and the resulting semantic graphs for both correct and incorrect documents are shown in Appendix Figures 1–3.
>
> This difference in semantic graph structure stems from the inherent characteristics of common attack methods. Although an adversary could create a poisoned document by simply flipping an edge or renaming an entity in Figure 2(a), such changes often reduce the attack's effectiveness or increase its cost. As a result, our defense approach, which takes advantage of these semantic and structural differences, proves particularly effective in such attack scenarios.
>
> ---
>
> **Q3. Line 192 – Source of verified documents**
>
> Thank you for this feedback.   The "Verifying Correct Documents" (D_cor) used for semantic reference are a small set of samples chosen manually from the dataset. These documents are only intended to provide the large model with accurate semantic structures for reference. To assess the impact of Dcor, we conducted ablation experiments that compared performance with and without this Dcor subset. As shown in the table below (with and without cor), enabling Dcor ("Y") consistently improves  accuracy or reduces ASR across PIA and different poisoning intensities (100%, 20%) on three datasets (Hotpot 、NQ、MS) .
>
> | W/O Dcor | PIA | 100%  | 20% | PIA | 100% |  20% |PIA | 100%  |20% |
> | --- | --- | --- | --- | --- | --- | --- | --- | --- | --- |
> | N  | 76 / 10 | 80 / 6  |  73 / 12 | 72  / 6 |63 / 3| 73 / 10 | 85 / 9|82 / 6 |  85 / 8  |
> | Y | 77.5 / 0.8 | 75.7 / 3.6  |72.7 / 4.5 |72.3 / 1.8 | 63.6 / 2.5  |74.5 / 10.2 |91.8 / 0 | 88.2 / 0 |89.1 / 9.1 |
>
> We will include representative examples in the appendix and investigate the effects of including or excluding this candidate set on the experimental results.
>
> ---
>
>
> **Q4. Equation (5) and Graph Input to LLM**
>
> Thank you for this feedback.   We appreciate the reviewer's observation that the description in lines 178-181 and Equation (5) is unclear.  To clarify, the input to the large language model is natural language not raw numerical vectors.  The compressed vectors are used solely for visualization and relationship construction purposes, but are not directly fed into the LLM.  Instead, we use EIRE outputs (e.g., structured triplets, triplets like "Entity A—[Relation R]→ Entity B") to construct semantic graphs using LLM. The LLM then uses these serialized representations to compare candidate and reference documents.We will update Equation (5) and the appendix text to reflect this process and avoid confusion about LLM input types.

---

> > ### Comment · Reviewer_N6z7 · 2025-08-06
> >
> > Thanks for the authors' rebuttal. However, I would like to follow-up on "Line 168 – Clustering of malicious documents in embedding space" and TrustRAG paper.
> >
> > First of all, I read the TrustRAG paper in more details and feel TrustRAG did a much better job detailing why malicious documents tend to form clusters in the embedded space. They closely relate this phenomenon to the actual attack method.
> >
> > Then I realize TrustRAG already has K-means based filtration and conflict resolution modules. So it seems the more evident novelty point in the current paper is the semantic graph filtration. However, in the ablation study, the authors pack k-means and semantic graph based filtration as SCF - so the actual contribution of semantic graph filtration is not clear enough. It is possible I miss the decoupled study of k-means based filtration vs. semantic graph based filtration. I would like more clarity on this part. Thanks again.

---

> ### Author Response · Authors · 2025-08-06
>
> Thank you for your comment. Different from previous work，our defense framework incorporating structured semantics into both the retrieval filtering and response generation stages of RAG. To our knowledge, this is the first paper to explicitly use semantic-level understanding via entity-intent-relation graphs and LLM-based validation to defend against poisoned attacks in RAG.
>
> This structured semantic filtering fills a critical gap in previous research, in which semantics were  ignored in both the filtering and inference stages of RAG defense. Unlike clustering-based defense methods, our semantic filter enables fine-grained detection of semantically inconsistent or poisoned content, thereby improving RAG's robustness.
> ﻿
>
> Our ablation results (below) show that the proposed semantic filter, combined with clustering filter, significantly reduces attack success rates while improving answer accuracy. Specifically, our hybrid method, Semantic-Clustering Filtering (SCF), is more robust than either component alone, demonstrating the complementary strengths of clustering and semantic reasoning.
>
> | Model | Module| 	PIA| 	100%| 	20%| 	PIA| 	100%| 	20%	| PIA| 	100%| 	20%|
> | --- | --- | --- | --- | --- | --- | --- | --- | --- | --- | --- |
> | Mistral | Clustering | 	78 / 5| 	81 / 2| 	78 / 9| 	68 / 3| 	65 /3 | 	70 / 10| 	85 / 7| 	82 / 7| 	82 / 12|
> | Mistral | Semantic 	| 79 / 4	| 80 / 2| 	74 / 11| 	69 / 2| 	64 / 3| 	73 / 8| 	86 /5	 | 82 / 6| 	86 /8|
> | Mistral | Both (SCF)	| 77.5 / 0.8	 | 75.7 / 3.6| 	72.7 / 4.5	| 72.3 / 1.8	 | 63.6 / 2.5 | 	74.5 / 10.2 | 	91.8 / 0 | 	88.2 / 0	| 89.1 / 9.1|
> | LLaMA | Clustering | 	73 / 6 | 	66 / 8 | 	66 / 18 | 	83 / 3| 	80 / 3  | 	82 / 8 | 	86 / 4 | 	88 / 3 | 	84 / 8|
> | LLaMA | Semantic | 73 / 4 	| 66 / 11 | 	66 / 19 | 	83 / 3 | 	79 / 4 | 	82 / 8 | 	87 / 4	 | 88 / 4 | 86 / 9 |
> | LLaMA | Both (SCF)	| 73.6 / 0.5	 | 72 / 10.9  | 	67.4 / 18.4	| 85.1 / 2.7 	 | 88.2 / 1.8  | 	86.9 / 4 | 	87.3 / 0.2 | 	 89.1 / 0	| 86.2 / 9.1|

---

> ### Author Response · Authors · 2025-08-08
>
> Dear Reviewer N6z7，
>
> Thank you for reading our response and submitting the acknowledgement. We would greatly appreciate it if you could also let us know whether our response has addressed your concerns. If so, we kindly ask you to consider increasing the score accordingly, if you haven’t already done so. Otherwise, if any questions or uncertainties remain, we would be more than happy to provide further clarification.
>
> Best, Authors

---

### Official Review · Reviewer_Djzt · 2025-07-03

**Clarity:** 3
**Significance:** 3
**Originality:** 2
**Rating:** 4
**Confidence:** 3

**Summary:**

The main objective of this paper is to identify and remove harmful documents from the knowledge base, thereby improving the reliability of the RAG system's output. First, this paper develops an EIRE method to identify the intent, entities, and relations in each document. Then, this papercombines clustering method with the EIRE method to eliminate harmful documents from the repository. Finally, after retrieval based on the filtered repository, this work further refines the retrieved documents using the EIRE method to ensure that only those documents that are both relevant to the query and factually consistent are retained as augmentations.

**Questions:**

1. In line 168 of the paper, it is mentioned that "Due to the semantic similarity of malicious documents, they cluster in embedding space." What is the basis or evidence supporting this claim?
2. Similarly, in line 188, it is stated that "poisoning documents generate sparse or fragmented graphs." While this is illustrated in Figure 2, what is the underlying reasoning behind this phenomenon?
3. In line 194 of the paper, it is mentioned that the similarity between the graph of a candidate document and some verified correct documents is used to evaluate the candidate document. Is the effectiveness of this operation supported by experiments?

**Ethical Concerns:**

["NO or VERY MINOR ethics concerns only"]

**Final Justification:**

I have thoroughly reviewed the authors' rebuttal and appreciate their comprehensive responses to my initial concerns, which strengthen the empirical foundation of their methodology. While these clarifications partially address the weaknesses noted, fundamental concerns regarding the inherent latency trade-offs (W3) and the explanatory depth of core mechanisms (W1, Q1-Q2) persist. Consequently, I will maintain the rating of Borderline Accept.

**Limitations:**

yes

**Quality:**

3

**Strengths And Weaknesses:**

Strengths
1. This work focuses on the task of removing corpus poisoning, which is highly important for improving the reliability of RAG systems.
2. This work employs a new approach, using EIRE to analyze and statistically evaluate documents at the semantic level to identify harmful ones.
3. The experiments in this work cover multiple large models and diverse datasets, with the results demonstrating the overall effectiveness of the proposed method.
Weaknesses
1. The explanation of the principles behind some key steps in this work's methodology is unclear, making it somewhat difficult to understand why this method is effective.
2. There is a lack of some key analysis experiments. For example, the paper mentions that SCF includes two filtering processes but does not employ experiments to explain why both must be used together.
3. The proposed method in this work introduces higher inference latency, which somewhat limits its practical applicability.

---

> ### Author Rebuttal · Authors · 2025-07-28
>
> We sincerely thank the reviewer for the thoughtful and constructive comments.  We address the remarks in Weaknesses (W) and Questions (Q) below.
>
> ---
>
> **W1. The explanation of the principles behind some key steps in this work's methodology is unclear, making it somewhat difficult to understand why this method is effective.**
>
> ---
>
> Thank you for this feedback. We clarify that the success of our strategy is derived from a two-stage semantic filtering framework: (1) To conservatively filter poisoned documents, SCF use both clustering and semantic graph matching (via output from our EIRE module); (2) CAF further filters retrieved content by ensuring semantic coherence between the query, retrieved documents, and the model’s internal knowledge. This layered design ensures robustness while minimizing information loss.  We have offered clarifications and examples in the appendix to assist the reader in understanding.
>
> ---
>
> **W2. Lack of ablation for the necessity of combining both SCF components**
>
> Thank you for this feedback. We agree that a more explicit evaluation of SCF's two subcomponents—the clustering-based and semantic graph-based filtering modules—is beneficial. While Figure 4 in Section 5.5 already shows an ablation study comparing SCF and CAF as a whole, it does not separate the contributions of the two SCF submodules.
> To address this, we carried out additional experiments, evaluating each subcomponent independently. The findings are summarized in the table below.
>
> |    Module   |      PIA   |      100%  |     20%     | PIA      |      100% |  20%     |    PIA   |     100%  |   20%    |
> | --- | --- | --- | --- | --- | --- | --- | --- | --- | --- |
> |   Clustering only    |   78 / 5    |  81 / 2     |  78 / 9     |   	68 / 3    |    65 /3	   |    70 / 10   |    	85 / 7		   | 	82 / 7	     |     	82 / 12  |
> |   Semantic only  |    79 / 4   |   80 / 2    |   74 / 11	    |   	69 / 2    |   64 / 3    |  73 / 8     |   86 /5    |  82 / 6	 |   86  /8   |
> |   Both (SCF)  |    77.5 / 0.8   |   75.7 / 3.6    |   72.7 / 4.5|   	72.3 / 1.8    |   63.6 / 2.5    |  74.5 / 10.2     |   91.8 / 0    |  88.2 / 0	 |   89.1 / 9.1    |
>
> Preliminary results show that combining both filters provides the best overall balance of clean accuracy and attack suppression. Specifically, while each module provides moderate improvements on its own, their combination results in significantly increased robustness (e.g., 0% ASR in several settings) with minimal effect on clean performance. This demonstrates the complementary nature of clustering and semantic-based filtering. These ablation results will be included in the revised appendix to support the empirical design choices underlying SCF.
>
> ---
>
> **W3. Inference latency may limit real-world applicability**
>
> Thank you for the feedback. We acknowledge that SeCon-RAG uses LLMs in several components (EIRE, semantic similarity scoring, and CAF's final judgment), resulting in a moderate runtime overhead. The need for deeper semantic understanding that conventional symbolic or sparse methods cannot offer drove our design decision to use LLMs for semantic structure extraction and graph similarity. We do agree, though, that this adds overhead. Also, we notice the following:
>
> - As shown in Section 5.4 and Figure 3, the additional runtime introduced by SeCon-RAG remains within a reasonable range (<1.5 minutes per batch). One query only requires an acceptable cost, which we believe is acceptable for many real-world applications that require robustness.
>
> - We also showed in Section 5.3 (Table 2) that SeCon-RAG performs well across various lightweight embedding models (e.g., MiniLM, SimCSE), indicating that the framework is adaptable to lower-cost alternatives. For future work, we are looking into more lightweight techniques to replace EIRE, reducing the reliance on full-scale LLM inference at runtime. We will clarify these points more explicitly in the revised version, as well as include additional discussion of scalability trade-offs in deployment scenarios.
>
> ---
>
> **Q1. Basis for the claim: "malicious documents cluster in embedding space" (line 168)**
>
> Thank you for the feedback. This insight is based on empirical observations that adversarially generated poisoning documents frequently use highly similar phrasing or templated structures, particularly when targeting the same query. Due to their similarities, they form tight clusters in the embedding space. This phenomena has previously been examined in past work, like as the paper "TrustRAG" which mentioned a similar viewpoint.
>
> ---
>
> **Q2. Reasoning behind: "poisoning documents generate sparse or fragmented graphs" (line 188)**
>
> Thank you for the feedback. This phenomenon arises for two reasons.
> - On the one hand, it is owing to the attack mechanism described in the paper, "poisoning RAG" settings, which require a lower cost to attack.
> - Poisonous content, on the other hand, frequently introduces isolated or deceptive claims that are not adequately supported by the surrounding semantic context. When EIRE extracts entities and their relationships from such documents, the resulting semantic graphs frequently include abrupt or unnatural relationships, as well as isolated nodes or disconnected subgraphs. This behavior differs from clean documents, which have well-structured, coherent semantic graphs. We demonstrate this qualitatively (Figure 2) and quantitatively (semantic similarity scores).
>
> We will elaborate on this reasoning and provide additional examples in the appendix.
>
> ---
>
> **Q3. Is there experimental support for semantic graph similarity filtering? (line 194)**
>
> Thank you for the feedback. To assess the impact of  "candidate document and verified correct documents" (D_cor) , we conducted ablation experiments that compared performance with and without this Dcor subset. As shown in the table below (with and without cor), enabling Dcor ("Y") consistently improves  accuracy or reduces ASR across PIA and different poisoning intensities (100%, 20%) on three datasets (Hotpot 、NQ、MS) .
> ﻿
> | W/O Dcor | PIA | 100%  | 20% | PIA | 100% |  20% |PIA | 100%  |20% |
> | --- | --- | --- | --- | --- | --- | --- | --- | --- | --- |
> | N  | 76 / 10 | 80 / 6  |  73 / 12 | 72  / 6 |63 / 3| 73 / 10 | 85 / 9| 82 / 6 |  85 / 8  |
> | Y | 77.5 / 0.8 | 75.7 / 3.6  |72.7 / 4.5 |72.3 / 1.8 | 63.6 / 2.5  |74.5 / 10.2 |91.8 / 0 | 88.2 / 0 |89.1 / 9.1 |
> ﻿
>
> We will include representative examples in the appendix and investigate the effects of including or excluding this candidate set on the experimental results.

---

### Note · Authors · 2025-08-12

We sincerely thank the AC, SAC, and reviewers for their careful reading, constructive feedback, and active engagement during the discussion phase.

**1. Core Novelty & Contributions**

To our knowledge, SeCon-RAG is the first RAG defense framework to incorporate structured semantics into retrieval filtering and response generation using the Entity-Intent-Relation Extractor (EIRE). This goes beyond prior works  by enabling fine-grained semantic graph analysis to detect poisoned or conflicting content, not only in retrieval but also in inference.Our two-stage design (SCF + CAF) jointly leverages statistical clustering and semantic graph filtering, then resolves  knowledge conflicts before answer generation.

**2.  Completeness**


We conducted extensive experiments using five different LLMs and three QA benchmarks in clean, high-poisoning, low-poisoning, and prompt-injection conditions. SeCon-RAG consistently had the lowest ASR and highest accuracy in nearly all settings, while maintaining clean-set performance. We further provided:

Decoupled Ablations distinguish between clustering and semantic filtering, demonstrating their complementary nature (e.g., Mistral-12B on HotpotQA: SCF alone ASR 0.8% vs. individual modules ≥4%).

Threshold Sensitivity remains stable with τ variations.

Latency analysis demonstrates acceptable overhead and suitability for lightweight embedding models.

**3. Addressing Reviewers' Concerns**

We clarified the foundation for  poisoned document clustering and sparse-semantic graph phenomena, including attack-mechanism reasoning, and  visualizations. We explained the verified documents (D_cor), as well as their measurable benefits through ablation experiment. We included a discussion of motivation and complexity, as well as a plan for revisions to improve clarity and scaleability.


**4. Reviewer Engagement & Outcome**

Two reviewers improved their scores following the rebuttal, and others' remaining concerns (e.g., SCF decoupling, novelty clarity) were addressed directly with new results and explanations. We believe that this fully resolves the issues raised.
﻿


We appreciate the opportunity to present this work, as well as the thoughtful feedback provided by the reviewers and AC, which helped improve the technical and empirical clarity of our paper.

---

### Decision · Program_Chairs · 2025-09-17

**Decision:**

Accept (poster)

**Comment:**

This paper introduces SeCon-RAG, a two-stage framework to enhance the trustworthiness of RAGsystems against corpus poisoning attacks. The core contribution is a novel defense mechanism that integrates structured semantic analysis into the filtering process. The first stage combines cluster-based filtering with a novel Entity-Intent-Relation Extractor (EIRE) to clean the retrieval database, while the second stage performs a conflict-aware check to ensure semantic consistency before generating a final answer.

The primary strengths of the work are its novel approach to the timely and critical problem of RAG security, along with its extensive and rigorous evaluation. The experiments are comprehensive, covering five different LLMs and three datasets, and the results demonstrate a consistent and significant improvement over baseline methods. The main weakness, identified by multiple reviewers, is the considerable inference latency and computational overhead introduced by the framework's reliance on multiple LLM calls for its core components, which may limit its real-world applicability.

During rebuttal, The authors provided a response that addressed most of the weaknesses raised. They conducted several new experiments, including detailed ablation studies to decouple the contributions of their filtering components, a standalone evaluation of the EIRE module, and a sensitivity analysis for the framework's thresholds . While they acknowledged the latency concerns, they justified it as an acceptable trade-off for achieving high robustness in critical applications. Overall, the paper's claims are well-justified, and the margin of improvement shown in the experiments is quite consistent. The rebuttal successfully answered most of the reviewers' questions, with two reviewers raising their scores in response. The main concern from the most critical reviewer was the novelty of the work in comparison to a prior paper, TrustRAG . The authors responded to this directly with a new, decoupled ablation study clearly demonstrating the added value of their semantic filtering module. As the reviewer did not reply further to this clarification, it appears the authors' response was sufficient. Given the strength of the paper and the comprehensive rebuttal, the paper is recommended for acceptance.